# Electromagnetic Radiation Effects on MgO-Based Magnetic Tunnel Junctions: A Review

**DOI:** 10.3390/molecules28104151

**Published:** 2023-05-17

**Authors:** Dereje Seifu, Qing Peng, Kit Sze, Jie Hou, Fei Gao, Yucheng Lan

**Affiliations:** 1Department of Physics and Engineering Physics, Morgan State University, Baltimore, MD 21251, USA; 2Physics Department, King Fahd University of Petroleum and Minerals, Dhahran 31261, Saudi Arabia; 3K. A. CARE Energy Research and Innovation Center at Dhahran, Dhahran 31261, Saudi Arabia; 4Hydrogen and Energy Storage Center, King Fahd University of Petroleum and Minerals, Dhahran 31261, Saudi Arabia; 5School of Materials Science and Engineering, Georgia Institute of Technology, Atlanta, GA 30332, USA; 6Nuclear Engineering and Radiological Sciences, University of Michigan, Ann Arbor, MI 48109, USA

**Keywords:** magnetic tunnel junction, irradiation, review

## Abstract

Magnetic tunnel junctions (MTJs) have been widely utilized in sensitive sensors, magnetic memory, and logic gates due to their tunneling magnetoresistance. Moreover, these MTJ devices have promising potential for renewable energy generation and storage. Compared with Si-based devices, MTJs are more tolerant to electromagnetic radiation. In this review, we summarize the functionalities of MgO-based MTJ devices under different electromagnetic irradiation environments, with a focus on gamma-ray radiation. We explore the effects of these radiation exposures on the MgO tunnel barriers, magnetic layers, and interfaces to understand the origin of their tolerance. This review enhances our knowledge of the radiation tolerance of MgO-based MTJs, improves the design of these MgO-based MTJ devices with better tolerances, and provides information to minimize the risks of irradiation under various irradiation environments. This review starts with an introduction to MTJs and irradiation backgrounds, followed by the fundamental properties of MTJ materials, such as the MgO barrier and magnetic layers. Then, we review and discuss the MTJ materials and devices’ radiation tolerances under different irradiation environments, including high-energy cosmic radiation, gamma-ray radiation, and lower-energy electromagnetic radiation (X-ray, UV–vis, infrared, microwave, and radiofrequency electromagnetic radiation). In conclusion, we summarize the radiation effects based on the published literature, which might benefit material design and protection.

## 1. Introduction

### 1.1. Tunnel Magnetoresistance

The phenomenon of tunnel magnetoresistance (TMR) has gained enormous attention in recent decades because of its essential applications in nonvolatile magnetoresistive random-access memory (RAM) and next-generation magnetic field sensors [1,2,3,4,5,6,7,8]. This interest follows the emergence and success of related magnetoresistance, such as anisotropic magnetoresistance (AMR) and giant magnetoresistance (GMR). Tunneling, as a foundational principle of TMR, arises from the quantum mechanical wave nature of particles and the non-zero probability of particles occupying classical forbidden regions.

The phenomenon of magnetoresistance (MR) was first discovered in 1856 [9] in nickel and iron sheets subjected to parallel or perpendicular magnetic fields; this is known as anisotropic magnetoresistance (AMR). The magnitude of electric resistance changed by about 2% at room temperature for alloy AMR materials [10]. The AMR effect was attributed to there being a higher probability of s−d scattering of electrons traveling along the direction of magnetic fields [11]. Since the 1970s, the AMR effect has been utilized for magnetic recording.

Subsequently, a significant resistance variation, up to 50%, was discovered in sandwich metallic magnetic Fe/Cr/Fe multilayers at room temperature in the late 1980s [12,13], known as giant magnetoresistance (GMR). GMR was characterized as the difference in electrical resistance between parallel magnetic states ($R_{P}$) and anti-parallel magnetic states ($R_{AP}$) normalized by the parallel resistance $R_{P}$:(1)$$MR=\frac{R_{AP}-R_{P}}{R_{P}}$$

The GMR effect has been attributed to the spin-dependent scattering occurring at interfaces [14]. Presently, GMR is being widely utilized in modern hard drives as a replacement for AMR devices for reading data.

Tunnel magnetoresistance (TMR) can be considered an extension of giant magnetoresistance (GMR) due to their similarities in electrical resistance changes in magnetic multilayer structures by aligning the magnetic moments of adjacent layers. Different from GMR, TMR employs a thin insulating layer as a tunneling barrier between magnetic layers, resulting in quantum mechanical electron tunneling across the barrier, which has a thickness of a few nanometers. This leads to more significant changes in electrical resistance compared to GMR devices.

TMR technology and devices emerged in the 1990s as a superior alternative to AMR and GMR devices for data storage due to their outstanding MR characteristics. Magnetic tunnel junctions (MTJs) are the core component of TMR devices. The development of MTJs was comprehensively reviewed recently [15,16,17]. Interested readers are encouraged to read the literature cited therein. Briefly, the tunneling magnetoresistance (TMR) effect, which is explained by spin-polarized tunneling electrons, was first observed in Fe/Ge-O/Co multilayers in 1975, with an MR ratio of 14% at 4.2K [18]. In 1994, amorphous aluminum oxide (Al${}_{2}$O${}_{3}$) was introduced as a tunneling barrier material, achieving MR ratios of 18% in Fe/Al${}_{2}$O${}_{3}$/Fe layers [19] and 70% in CoFeB/Al${}_{2}$O${}_{3}$/CoFeB structures [20,21] in the 2000s. AlO${}_{x}$-based MTJs have been reviewed recently and interested readers are referred to the literature listed in [22]. MgO-based MTJs were first investigated in the 1990s [23]. A moderate TMR of 20% was achieved at room temperature in amorphous MgO-based MTJs. Crystalline MgO was later utilized as a tunneling barrier material, resulting in room temperature MR ratios of 30% [5], 67% [24], 88% [25], 180% [2], and 220% [26] in crystalline Fe(001)/MgO(001)/Fe(001) MTJs, 230% [3] and 355% [27] in CoFeB/MgO/CoFeB MTJs, 410% in Co(001)/MgO(001)/Co(001) MTJs [28], and 500% [15,16,29] and 604% [30] in CoFeB/MgO/CoFeB MTJs. The highest MR ratio, 1144%, was observed at 4.2K in CoFeB/MgO/CoFeB MTJs [30]. The MR ratios of MgO-based MTJs have increased by over 50 times in less than two decades since the initial report, with some reviews of giant TMR in MgO-based MTJs published [15,31]. The history of the MR ratios of both AlO${}_{x}$-based and MgO-based MTJs is plotted in Figure 1. Clarifying the radiation tolerance of these devices will lead to a deeper understanding of the physics of spin-dependent tunneling states.

MTJs consist of two ferromagnetic layers separated by a very thin insulator with a nanometer-scale thickness, typically made of amorphous Al${}_{3}$O${}_{3}$ or crystalline MgO. Figure 2 shows a typical MTJ consisting of a MgO crystalline barrier and Fe layers. Electrons tunnel across the insulating nanolayer from one ferromagnetic layer to the other, thereby contributing to the junction’s electric conduction. The resistance of an MTJ is dependent on the relative magnetic alignment, either parallel or anti-parallel, of its ferromagnetic layers and its thin insulating layer.

Various ferromagnetic materials, including Fe, Co, FeCo alloys, and FeCoB have been employed as MTJ ferromagnetic layers. Typically, these layers are crystalline in nature, with a specific orientation, such as Fe(001), chosen to match the crystalline barrier and increase the MR ratio.

### 1.2. Applications of MTJs

MTJs have a broad range of applications in electronics, sensing, energy generation, and energy storage owing to their unique tunneling properties. A brief overview of these applications is presented below. Since the MR of MgO-based MTJs is significantly higher than that of AlO${}_{x}$-based MTJs, our focus will only be on MgO-based MTJs here.

#### 1.2.1. Electronics

One of the most well-known applications of MTJs is their use in data storage, particularly in MTJ-based memory devices [17,32,33,34], including dynamic random-access memory, flash memory, and hard disk drives. Data can be stored without the need for external magnetic fields [35]. Many review articles have been published on this topic, as listed in the preceding section. MgO-based TMJs exhibit high MR ratios at room temperature and have been utilized in hard disk drives (HDD) with a high density [17,33,36,37,38,39,40].

Additionally, an MTJ is comprised of two distinct states, namely, parallel and anti-parallel. Consequently, a single MTJ has the capability to store data in four different states [41]. Therefore, stacked MTJs are suitable for use in magnetic random-access memory (MRAM) applications [2,7,17,31,42,43,44,45,46,47,48,49,50]. These nonvolatile MRAMs demand high MR ratios of >150% at room temperature.

Figure 3 shows a typical structure of an MRAM element. The element’s primary component is an MTJ, which comprises a ferromagnetic free layer, an insulating tunnel barrier, and a ferromagnetic fixed layer. These sandwiched layers exhibit TMRs as a result of spin-dependent electron tunneling [31,42,43,51]. A recent review of the structure of MRAMs can be found in reference [49].

MRAM has the potential to replace all existing memory devices because of it capability of combining the speed of static random-access memory (SRAM) and the density of dynamic random-access memory (DRAM), while also being nonvolatile like hard disk drives (HDD). Therefore, MRAM is a highly desirable form of memory [16,20,32]. As a result, MTJ-based MRAM devices have been extensively investigated in the past two decades.

Compared with other kinds of random-access memory, MTJ-based MRAMs are a type of nonvolatile memory that can work in irradiation environments, such as in outer space applications [52]. The radiation tolerance of these MRAMs is critical to their effectiveness in such harsh environments.

In addition, MTJs have the potential to be employed as magneto-electric spin logic devices, which are capable of converting analog signals to digital ones. Various designs of analog-to-digital converters (ADCs) have been proposed [53,54,55], including sigma–delta (Σ−Δ) ADCs with high bit resolution [55]. As compared to traditional ADCs, the energy consumption of these MTJ-based ADCs is very low, down to 66fJ for 4-bit MTJ-based ADCs and 37fJ for 3-bit MTJ-based ADCs [54].

MTJ devices also have significant potential for sensing, such as ultra-sensitive magnetic field sensors [56,57], microwave frequency sensors [58], microwave power sensors [59], thermal sensors [60], and heat sensors [61].

#### 1.2.2. Energy Harvesting

In addition to their conventional applications in memory and sensors, MTJ devices hold promise for renewable energy generation and storage. While relatively new in the field, MTJ-based energy devices have attracted considerable research attention. Although their device efficiencies are lower than those of traditional energy devices, the novel heterostructures of MTJs have the potential to significantly impact the area. Many research groups have explored the fundamentals and future prospects of energy applications involving the spin of electrons. Various kinds of energy, ranging from heat and light to mechanical vibrations, have been successfully converted to electricity through spin conversion [34,61,62].


*Heat:*


Based on EU data, a considerable amount of industrial energy consumption, ranging from 20% to 50%, is lost as waste heat. In the United States, up to 1734 trillion Btu of waste heat went unrecovered in 2008 [63]. The Seebeck effect, which involves electromotive force (emf) generation under a temperature gradient, has been widely investigated in the past decades. Thermoelectric (TE) devices have the capability to convert heat to electricity based on the Seebeck effect. Because of their unique characteristics, such as having no moving parts, quiet operation, a low environmental impact, and high reliability, TE devices have attracted widespread interest since their discovery. In the past decades, semiconductor TE materials, especially ceramic nanocomposite bulks [64,65,66,67,68], have been developed for this purpose. Up to now, various TE nanocomposites have been investigated in the absence of magnetic fields [69,70,71,72,73,74,75,76,77,78,79,80]. Device efficiencies of up to 10% can be achieved using semiconductor TE devices.

Spin caloritronics, the combination of spintronics and thermoelectrics, is an emerging field [61,81]. Electron spin waves interact with heat in insulating ferromagnets under magnetic fields through the magneto-Seebeck effect, also referred as the spin Seebeck effect or magneto-thermopower effect. A thermal gradient can lead to the production of magneto-thermopower and magneto-thermocurrent [82]. Therefore, spin caloritronic devices can serve as waste heat recyclers and heat sensors under magnetic fields.

MTJ devices, comprised of insulating barriers and ferromagnetic layers, can utilize spin caloritronics to generate pure spin currents via magnetization dynamics induced by a temperature gradient. These MTJ devices have a unique potential in harvesting thermal energy and there have been many studies focused on MTJ-based heat recycling in the past decade. The spin Seebeck coefficients of various MTJs have been measured under magnetic fields [81,82,83,84,85,86,87]. For example, CoFeB/MgO/CoFeB MTJs have been integrated with resistance thermometers to recycle waste heat from the spin Seebeck effect [61]. A Seebeck coefficient of Al${}_{2}$O)3-based MTJs was measured up to 1mV/K [88]. A large spin-dependent Seebeck coefficient of 100μV/K was observed in CoFeB/MgO/CoFeB MTJs [89]. However, due to their nanoscale thickness, the output power of MTJs is much lower than that of their semiconductor TE bulk counterparts (up to kW). It was reported that the output TE power of a CoFeB/MgO/CoFeB MTJ device was only 10pW per $12.6\;{\mathrm{cm}}^{2}$ (∼$10\;nW/m^{2}$) [90]. Even compared with that of semiconductor TE film devices (up to several hundred W/m${}^{2}$), the output power is very low for the state-of-art MTJ devices. Although the power output of present MTJ devices is unsatisfactorily low for industrial heat recyclers, MTJ devices are one kind of emerging energy-harvesting device.


*Solar energy:*


In addition to their capacity for heat recycling, MTJs can generate electricity through the utilization of solar energy. A phonon can couple to the electron spin and magnon, which enables the generation of spin currents from solar energy [34]. More recently, photoinduced spin currents were observed [62]. Furthermore, the potential of MTJs was explored for spin photovoltaic applications [91].


*Mechanical energy:*


Recently, a new research field known as spinmechanics or spinmechatronics has emerged, combining spin currents with mechanical motion [62]. Spin currents can be generated from mechanical energy such as vibrations and sounds [92,93].

In brief, MTJs have the capacity to convert different kinds of energy into electricity through the amalgamation of electron spin with established energy conversion techniques. These research areas are relatively nascent and are expected to find many applications in the forthcoming decades.


*Electromagnetic energy:*


It was reported that MgO-based MTJs could produce significant DC voltage when exposed to microwave radiation [94]. A DC voltage was generated under microwave irradiation with a frequency of 1MHz to 40GHz and power density of 10–10 × 10${}^{6}$ mW/m${}^{2}$, with a sensitivity of up to 5000 mV/mW. A similar phenomenon was also observed in AlO${}_{x}$-based MTJs exposed to microwave radiation with a power of 1–100 mW and frequency of 1.5–2.5 GHz [58].

#### 1.2.3. Energy Storage

MTJs have potential applications in the field of energy storage as well, particularly with respect to batteries and capacitors, which are two kinds of popular devices to store energy. Recent work on MTJ-based energy storage devices is highlighted below.


*Capacitors:*


The magnetic capacitance of MTJs was first investigated in Co/Al${}_{2}$O${}_{3}$/Co MTJs in the 2000s, and their potential application as supercapacitors for energy storage was explored [95]. The tunneling magnetocapacitance (TMC) of Co${}_{40}$Fe${}_{40}$B${}_{20}$/MgO/Co${}_{40}$Fe${}_{40}$B${}_{2}$ MTJs was measured at room temperature about two decades later [96,97]. The voltage-induced TMC ratio reached 1000% due to the emergence of spin capacitance. An inverse of TMC was observed in Fe/AlO${}_{x}$/Fe${}_{3}$O${}_{4}$ MTJs [98]. The inverse TMC reached up to 11.4% at room temperature and could potentially reach 150%. It is believed that the spin accumulation in anti-parallel configurations of MTJs leads to a difference in spin-up and spin-down diffusion lengths, creating a charge dipole that acts as an extra serial capacitance and gives rise to the observed TMC effect [99].

In a recent study, it was reported that MgO-based (001)-textured MTJs exhibited a significant TMC, of 332% at room temperature [100]. Subsequently, an even higher TMC, of over 420% at room temperature, was achieved using epitaxial MTJs with MgAl${}_{2}$O${}_{4}$ (001) barriers possessing a cation-disordered spinel structure [100].

There findings highlight the potential of MTJs in the development of capacitors and related technologies.


*Batteries:*


MTJ devices have also been employed as spin batteries for the conversion of the magnetic energy of superparamagnetic nanomagnets into electricity [101]. The examined MTJs contained MnAs nanomagnets with a zinc-blende structure. These nanomagnets were chargeable under magnetic fields, providing evidence for the existence of spin batteries. The resulting electromotive force (emf) was found to operate on a timescale of approximately $10^{2}$–$10^{3}$ s. The emf should result from the conversion of the magnetic energy of the superparamagnetic nanomagnets into electrical energy during their magnetic quantum tunneling.

MTJ devices have diverse applications, such as data storage, sensors, energy generation and storage, even under irradiation. Consequently, it is crucial to evaluate their capacity to withstand irradiation. Here, we focus on the radiation tolerance of MgO-based MTJs. This information provides valuable insights into their stability, and benefit error-free operation and protection of MgO-based MTJ devices in irradiation environments.

### 1.3. Irradiation

MgO-based MTJs may work in various irradiation environments. Therefore, it is necessary to review both natural and artificial sources of radiation prior to reviewing the radiation tolerance of MgO-based MTJs.

#### 1.3.1. Natural Radiation Sources

The Sun is the major natural radiation source in our life [102,103]. Nuclear fusion processes within the Sun produce cosmic rays that consist of high-energy atomic nuclei and electromagnetic waves, which spread through the solar system. These primary cosmic rays are composed primarily of 99% nuclei (protons accounting for 90%, alpha particles accounting for 9%, and heavier element nuclei making up 1%) and approximately 1% solitary electrons and the electromagnetic component (gamma rays, X-rays, UV–visible light, and IR light). The energy of the primary cosmic rays is high, up to $10^{20}\;\mathrm{eV}$. Owing to the Earth’s magnetic field, the energetic particles are deflected and trapped within the Van Allen radiation belts. The belts extend from an altitude of about 640km to 58,000 km above the Earth’s surface, as shown in Figure 4a. Various spacecraft components, including MTJ devices, can be exposed to primary cosmic rays in the Van Allen belts.

Upon entering the Earth’s atmosphere, the primary cosmic rays collide with atoms and molecules present in the atmospheric layers [104]. These collisions produce secondary cosmic particles with lower energy and electromagnetic waves. The secondary particles and electromagnetic waves include low-energy neutrons, protons, electrons, alpha particles, γ-rays, and X-rays. The energy of the secondary cosmic particles and electromagnetic waves is much lower than that of the primary cosmic rays, but is still considerable. For instance, the energy of the secondary γ-rays can be 50MeV on the Earth. Due to their sufficiently high energy, these secondary cosmic particles and electromagnetic waves can potentially damage MTJ devices, leading to soft errors in MTJ-based electronic integrated circuits.

There are natural radioactive minerals on the Earth, such as compounds containing uranium-238 (U-238) and thorium-232 (Th-232) radionuclides. These radioactive elements emit high-energy particles or rays in the natural environment. As such, these minerals are another kind of natural radiation source on the Earth.

Figure 4b shows the full spectrum of electromagnetic radiation on the Earth. Table 1 lists the wavelength, frequency, and energy of various types of electromagnetic wave.

Thus, it is necessary to investigate the potential radiation effects on microelectronic devices that are exposed in outer space or on the Earth. This is particularly important in the case of MTJ devices that are deployed in spacecraft, satellites, and airplanes, which operate in an irradiation environment filled with high-energy particles and high-energy electromagnetic waves.

#### 1.3.2. Artificial Radiation Sources

Besides the natural radiation sources, various artificial sources of radiation exist on the Earth, including nuclear weapons, nuclear power plants, television transmitting towers, microwave ovens, and wireless phones. These artificial radiation sources are also omnipresent in our surroundings, as shown in Figure 4b. For instance, modern microwave ovens used in kitchens can produce microwaves with a frequency of 2450MHz [106]. Cellphone towers can emit electromagnetic radiation with frequencies of 800MHz and 1900MHz for 3G cellphone communications [107,108], with frequencies of 24–47 GHz for high-band 5G phones [108,109]. Moreover, even human bodies can emit infrared radiation [110]. Although the energy of this artificial radiation is significantly lower than that of cosmic rays, it is still required to know if this artificial radiation damages MTJ devices or degrades MTJ device performance. Therefore, this review paper comprehensively examines the radiation impacts of various electromagnetic waves, including γ-rays, X-rays, UV–visible light, microwaves, and even infrared radiation.

To date, various artificial radiation sources have been utilized in laboratories to quantitatively investigate the radiation effects on MgO-based MTJ devices. Most of the data reviewed here were collected using these radiation sources. Table 2 lists some typical radiation sources utilized in the cited literature here.

These artificial radiation sources can produce controllable electromagnetic particles in laboratories. Particle accelerators and synchrotron radiation, for instance, can generate high-energy particles, including neutrons and electrons, with energies ranging from 0.1MeV to 1.0MeV and a high flux. γ-rays are usually generated from radioisotopes in laboratories, with energies from several keV to MeV. Some specialized devices, such as electron microscopes, can produce middle-energy particles, of 5–200 keV. Commercial X-ray tubes can emit low-energy X-rays, of tens of electronvolts. Various light sources, such as xenon and halogen bulbs, can generate UV–visible light with energies in the electronvolt range. Additionally, infrared radiation below 1eV can be generated from electric furnaces in laboratories.

#### 1.3.3. Radiation Units

The impacts of radiation are generally categorized into three types [103,104,122]: (1) total ionizing dose (TID), which is quantified in rad or gray units. TID effects can change the threshold voltages of electronic devices due to trapping of charges during radiation exposure. TID may cause leakages of electric currents. (2) Single event effects (SEE), which are not cumulative but result from individual interactions. SEE may cause soft errors and hard errors of devices. (3) Displacement damage dose (DDD), which can generate lattice defects. Sufficient displacement may change the device or material’s performance properties over time. TID and SEE are examples of ionizing radiation effects, while DDD is an instance of a non-ionizing radiation effect. TID and DDD can lead to lasting damage to electronics over an extended period, showing long-term effects, whereas SEE typically results in immediate short-term effects. However, both short-term and long-term effects can potentially have permanent consequences.

To facilitate comprehension of the impact units, a brief summary is provided here. There are four kinds of ionizing radiation quantities: (1) Activity quantity, with units of becquerel (Bq), curie (Ci), and rutherford (Rd); (2) exposure quantity, with units of coulomb per kilogram (C/kg) and röntgen (R); (3) absorbed dose quantity, with units of gray (Gy), erg per gram, and radiation absorbed dose (rad); and (4) equivalent dose quantity, with units of sievert (Sv) and röntgen equivalent man (rem). The definitions of these radiation quantities are also listed in Table 3 for readers without a background in radiation.

### 1.4. Properties of MTJ Materials

MgO-based MTJs are composited of MgO insulating barriers and ferromagnetic layers. The ferromagnetic layers consist of free layers and fixed layers, typically made of ferromagnetic Fe and CoFeB. In order to understand the radiation tolerance of MgO-based MTJs, the physical properties of MgO and Fe/COFeB are first summarized below. The radiation tolerance of MgO-based MTJs is related to these properties.

#### 1.4.1. Magnesium Oxide Barrier

Magnesium oxide (MgO) possesses an ionic bonding structure, consisting of Mg${}^{2+}$ and O${}^{2-}$, with a crystallographic structure of rock salt (NaCl). Figure 5 shows its crystallographic structure. Figure 6 shows its monolayer structure. Magnesium and oxygen atoms alternately stack in the lattice.

MgO is an excellent electrical insulator, exhibiting a conductivity of $10^{-14}\;\mu S/m$ at room temperature. Additionally, it is a soft magnetic material, with a magnetic susceptibility of $-10.2\times 10^{-6}\;{\mathrm{cm}}^{3}/\mathrm{mol}$. The compound is also a refractory material, with physical and chemical stability up to $2500{\;}^{\circ }C$. Its physical properties are listed in Table 4.

#### 1.4.2. Ferromagnetic Layers

Ferromagnetic materials are utilized as free/fixed layers in MgO-based MTJs. Crystalline (001) iron films were initially used as free/fixed layers in MgO-based MTJs to achieve an MR ratio of 220% [26,124] at the beginning of the 2000s. Subsequently, crystalline Co(001) films were employed as free/fixed layers of MgO-based MTJs, achieving an MR ratio of 410% [28]. Currently, CoFeB is extensively used in MgO-based MTJs, and the MR ratio has been enhanced to 500–600% at room temperature [15,16,30]. The structural, thermal, and magnetic properties of these three ferromagnetic materials are listed in Table 5.

### 1.5. Theoretical Radiation Tolerance of MTJs

Radiation-induced damage to electronic circuits has been known since the 1950s. In the 1970s, memory and logic perturbations were detected in satellite electronic devices as a result of heavy-ion radiation within the solar wind [129]. Subsequently, soft errors caused by cosmic rays were reported in Si-based DRAM memory chips at the end of the 1990s [130]. Serving as a counterpart to Si-based devices, the stability of MTJ devices has also been investigated. In this subsection, the theoretical work will be discussed, while the experimental research will be covered in the subsequent section.

Theoretical investigations of the radiation effects on MTJs were initially carried out using the Julliére model [18] and the theory of electron tunneling, both of which established TMR models. In this subsection, the Julliére model will be first discussed, followed by the electron tunneling model.

According to a report in 1997 [131], the Julliére model is more suitable for amorphous barriers, not a precise representation of the magnetoconductance exhibited by free electrons tunneling through a crystalline barrier. Instead, in the case of thick barriers, Slonczewski’s model may offer a more accurate approximation. Ionizing radiation, such as γ-rays, can displace atoms and create local lattice disorder, leading to the formation of amorphous regions in barrier layers. Therefore, despite this limitation, the Julliére model is employed here to illustrate the effect of an amorphous state in barrier layers, which is induced by irradiation. The model offers a simplified visual representation of the degradation caused by irradiation.

In non-magnetic materials, the populations of spin-up electrons and spin-down electrons are equal, and are randomly distributed in an equilibrium state. Conversely, in ferromagnetic materials, electron spins are aligned spontaneously, resulting in unequal numbers of spin-up and spin-down electrons. The unequal spin-up and spin-down electrons can tunnel into the empty states of the initial spin channel, which affects electrical resistance under magnetic fields, resulting in non-zero MR ratios. The MR ratios of an MTJ can be expressed in terms of the conduction electron spin polarization $P_{i}$ of the ferromagnetic layers [18,132].
(2)$$\mathrm{TMR}\;\mathrm{Ratio}=\frac{2P_{1}P_{2}}{1-P_{1}P_{2}}$$
where
(3)$$P_{i}=\frac{D_{i,↑}\left(E_{F}\right)-D_{i,↓}\left(E_{F}\right)}{D_{i,↑}\left(E_{F}\right)+D_{i,↓}\left(E_{F}\right)}$$

Here, i=1,2. $D_{i,↑}\left(E_{F}\right)$ and $D_{i,↓}\left(E_{F}\right)$ are the spin-dependent densities of states of the free/fixed layers at the Fermi energy ($E_{F}$) for the majority-spin and minority-spin bands. The spin polarization of the free/fixed layers $P_{i}$ (i=1,2) is affected by the free/fixed layer materials. Based on the Julliére model, any factors changing the Bloch states (such as momentum and coherency) within the free/fixed layer can affect the tunneling probabilities and change the TMR ratios.

The concept of electron tunneling can explain MTJ too, with a particular focus on crystalline barrier MgO-based MTJs [133,134]. It is generally accepted that the effectiveness of MgO-based MTJs is highly dependent on the crystallinity of the insulating MgO barrier.

Figure 7 schematically illustrates coherent tunneling transport in MgO(001)-based MTJs. As illustrated in the schematic, there are three kinds of evanescent states (also known as tunneling states) for ideal coherent tunneling in the band-gap of MgO(001): $\Delta_{1}$, $\Delta_{2}$, and $\Delta_{5}$. $\Delta_{1}$ Bloch states are highly spin polarized in the ferromagnetic layers, and tunneling probability is a function of $\kappa_{\|}$ wave vectors. Theoretical studies suggested that the ferromagnetic $\Delta_{1}$ states dominate the tunneling process through the MgO(001) barrier [133,134]. When the symmetries of tunneling wave functions are conserved, ferromagnetic $\Delta_{1}$ Bloch states can couple with MgO $\Delta_{1}$ evanescent states, which have the slowest decay and highest tunneling probability [133] along the [001] direction. The dominant tunneling channel for the parallel magnetic state is free layer $\Delta_{1}↔$ MgO $\Delta_{1}↔$ fixed layer $\Delta_{1}$. In the parallel magnetic states, the majority-spin conductance occurs dominantly at $\kappa_{\|}=0$ because of the coherent tunneling of majority-spin $\Delta_{1}$ states. In contrast, for the minority-spin conductance in the parallel magnetic state and the conductance in the anti-parallel magnetic state, spikes of tunneling probability would appear at the finite $\kappa_{\|}$ points. Although a finite tunneling current flows in the anti-parallel magnetic state, the tunneling conductance of the parallel magnetic state is much higher than that in the anti-parallel magnetic state, leading to a very high MR ratio.

According to the theory of electron tunneling, any modification to the symmetry of the MgO barriers and ferromagnetic free/fixed layers would affect the MR ratio of MgO-based MTJs. This means that the symmetry of both the propagating states in the magnetic layers and the evanescent state in the MgO barrier is critical in determining the tunneling conductance. The symmetry matching of the Bloch actively controls the tunneling conductance and MR states in both the free/fixed layers and the evanescent states in the barrier. Any changes to the symmetry of the MgO barrier and magnetic layers would affect the effective $\Delta_{1}$ states between the MgO barrier and ferromagnetic layers, thereby changing the MR ratios.

As discussed above, several essential factors, including the crystallinity and crystallographic orientation of both the barrier and ferromagnetic layers, play essential roles in the MR ratio. The presence of disorders, such as surface roughness, interface inter-diffusion, and impurities, as well as defects such as grain boundaries, stacking faults, and vacancies, would significantly affect the spin polarization and tunneling conductance.

Irradiation is a source of defects in MTJs and potentially affects the MR effects. Various types of ionizing radiation, such as α-particles, β-particles, and high-energy ions, as well as non-ionizing radiation, including neutrons, electromagnetic radiation such as γ-rays and X-rays, and thermal radiation, could degrade MR performance if any microstructures of MTJs are modified.

The radiation tolerance of AMR and GMR sensors has been experimentally investigated [135,136,137]. It was found that these sensors are generally somewhat resistant to radiation. The radiation tolerance of MTJ devices has also been experimentally studied. It was believed that polarization of the conduction currents and MR ratios of MTJs would be reduced if the interfaces between the tunneling oxide barrier and the ferromagnetic layers were damaged by radiation, which results in spin scattering defects [138]. Any permanent damage to the oxide barrier, usually caused by high-energy radiation, would cause leakage paths and reduce the tunneling resistance of MTJs. Low-energy radiation would cause cumulative degradation of MTJs.

A recent review analyzed the effects of radiation on Al${}_{2}$O${}_{3}$-based MTJs [139]. High-energy heavy-ion irradiation usually caused the most displacement damage in this kind of MTJ, leading to a deterioration of magnetotransport properties with increasing radiation dose. High-energy protons and γ-ray radiation have minimal effects on the magnetic properties of AlO-based MTJs, suggesting that AlO-based MTJs may be promising candidates for radiation applications.

Compared with the oxide barriers in AlO-based MTJs, the MgO barriers in MgO-based MTJs are thinner, usually 1–2 nm thick. The thinner crystalline layers would be more sensitive to irradiation, as observed in other two-dimensional materials [112,140], affecting the performance of MgO-based MTJs significantly.

Here, we review the literature on the effects of irradiation on MgO-based MTJs, summarize the published experimental data, and evaluate the resulting irradiation effects. This review will highlight the state-of-the-art findings of the effects of electromagnetic radiation on MTJs with MgO barriers.

## 2. Effects of Cosmic Radiation

Primary cosmic rays and secondary high-energy cosmic rays include high-energy protons, alpha particles, nuclei, electrons, and various electromagnetic waves. Cosmic rays can be classified into four types: heavy ions, mid-mass subatomic particles (proton and neutron), light-mass subatomic particles (electron), and massless electromagnetic waves. The effects of the first three kinds of cosmic radiation are briefly reviewed in this section. The effects of electromagnetic radiation will be reviewed in the subsequent section.

At sea level, the average annual cosmic ray dose is about 0.27mSv (27mrem). The radiation dose is about 0.10μSv/h. The cosmic radiation dose increases rapidly with increasing altitude, reaching about 2.0μSv/h at 9km altitude and about 9μSv/h at 18km above the Earth’s surface. Therefore, it is necessary to examine the effect of cosmic radiation on TMR-based MTJs utilized in daily life, especially in spacecraft and satellites. It is generally accepted that the high-energy particle radiation, such as high-energy ions, neutrons, protons, and electrons, can degrade the performance of MgO-based MTJ devices.

### 2.1. High-Energy Heavy-Ion Irradiation

Insulating oxide barriers can be degraded by heavy-ion radiation. It was reported that ultra-thin aluminum oxide tunnel barriers were damaged by highly-charged ions (such as Xe ions with 19–42 keV) [141,141]). The conductance of AlO-based MTJs linearly increases with radiation flux [113]. Furthermore, high-energy light ions (such as carbon and oxygen ions) and heavy ions (such as nickel ions) within 10MeV decreased the MR ratio of AlO-based MTJs irreversibly as the ion flux increased [142].

MTJs’ MgO dielectric barriers are susceptible to radiation too. Typically, ionizing radiation usually generates charge trap centers in MgO barriers and the interfaces between MgO barriers and ferromagnetic layers. The produced charge trap centers can lead to extra noise of MTJs [143] and reduce the MR ratios of MTJs [144] by perturbing tunneling processes.

MTJ ferromagnetic materials are also susceptible to radiation. It has been well known for decades that ion radiation can damage the crystallographic structures of MTJ ferromagnetic layer materials and change their physical properties [145]. Generally, ion radiation would cause displacement damage, which affects the microstructure and properties linked to displacement damage [139]. It was reported that high-energy argon ions, with energies of 44MeV, and krypton ions, with energies of 35MeV, created amorphous zones [146] or defects [147] in BaFe${}_{12}$O${}_{19}$ magnetic materials, changing their magnetic properties and microstructures. High-energy helium ion radiation can create He nanobubbles at ion implantation regions [148] and induce up to a 36% change in the crystal anisotropy [149] of ferroelectric LiNbO${}_{3}$ materials.

The radiation-induced damage of oxide barrier materials and ferromagnetic layer materials would affect the behavior of MTJ devices. It was reported that CoFeB/MgO/CoFeB MTJs were degraded by high-energy oxygen ion (O${}^{-}$) radiation during RF sputtering [150]. Table 6 lists some ion irradiation effects on MgO-based MTJs. It is generally accepted that high-energy irradiation usually degrades the TMR behavior of MgO-based MTJs.

It was also reported that MgO-based MTJ devices exhibit radiation tolerance. NASA conducted a test of an MTJ-based MRAM (MR2A16A from Freescale Semiconductor Inc.) under a heavy ion single event [111]. The tested MRAM was exposed to 3.0GeV Kr ions, 1.6GeV Ar ions, and 3.2GeV Xe ions. The test results indicated that the MRAM device was sensitive to single-event latchup (SEL), which was attributed to the complementary metal–oxide–semiconductor (CMOS) portion of the device. However, there was no indication that MTJ elements were damaged from heavy ions.

Radiation tolerance of MTJ devices was also reported by other research groups. Kobayashi et al. exposed CoFeB/MgO/CoFeB MTJs to high-energy Si ion radiation with energies of 15MeV[114]. The MTJs (consisting of Mg(1.3nm) and CoFeB (1.5nm) were sandwiched between 200μm additional electrodes. Only minimal degradation (∼1%) was observed in their electrical resistance. However, no significant changes were detected in the retention states before and after the irradiation.

### 2.2. High-Energy Proton Irradiation

Hughes et al. irradiated MgO-based MTJ devices (MRAM) utilizing proton ions with energies up to 220MeV and doses up to $1\times 10^{12}\;\mathrm{proton}/m^{2}$ [119]. The MTJ devices consisted of Ru(7 nm)/Cu(20 nm)/Ta(5 nm)/CoFeB(2.2 nm)/MgO(1.2 nm)/CoFeB(2.5 nm)/ Ru(1 nm)/CoFe(2.5 nm)/PtMn(15 nm)/Ta(0.5 nm)/Cu(100 nm)/Ta(0.5 nm)/SiO${}_{2}$(100 nm)/Si (substrate). The magnetization, ferromagnetic resonance, and tunnel magnetoresistance were examined before and after proton exposures. No changes were observed in their material properties. No radiation effects were observed.

Snoeck et al. exposed Au(∼10 nm)/Pd(∼20 nm/Fe(30 nm)/MgO(0.6 nm)/Fe(10 nm) magnetic tunnel junctions under 150keV nitrogen ions (N${}^{+}$) at a flux of $5\times 10^{15}\;\mathrm{ions}/{\mathrm{cm}}^{2}$ and $3\times 10^{16}\;\mathrm{ions}/{\mathrm{cm}}^{2}$ [153]. Bi-linear and bi-quadratic coupling increased gradually with increasing ion dose. However, no complete description of the irradiation-induced effects was reported.

### 2.3. High-Energy Neutron Irradiation

High-energy neutron irradiation usually alters atomic arrangements and damages crystalline structures of many materials. The radiation can also create nanoscale amorphous regions within crystal lattices [154]. While metals are relatively immune to ionizing radiation due to their ionic bonds, fast neutrons can still enter metals and cause significant structural damage. For instance, neutron-radiation-induced defect clusters and cavities in copper [155], decreased magnetic remanences of NdFeB permanent magnets [115], and changed the Curie temperature of FeNiCrMoSiB amorphous alloys [156].

High-energy neutron irradiation can also damage the ferromagnetic layers of MgO-based MTJs. High-energy neutrons can travel in the crystalline lattice of free/fixed layers and displace these atoms from their initial lattice positions through kinetic energy transfer. These kind of displaced atoms are termed primary knock-on atoms (PKAs). The PKAs can continuously displace other lattice atoms that are named secondary knock-on atoms (SKAs). This series of displacements can generate numerous defects in the crystalline free/fixed layers, ultimately affecting the performance of TMJs. Table 6 lists one case of neutron irradiation, which is generally understood to degrade MTJ devices.

### 2.4. High-Energy Electron Irradiation

High-energy electron irradiation affects MTJ component materials. In one study, amorphous CoFeB thin films (which are used as free/fixed layers of TMJs) were exposed to an electron beam with an energy of 200keV in a transmission electron microscope [116]. The electron radiation modified the phase and microstructure of the films. Another study examined the thermoluminescent properties of ultrafine MgO particles, with sizes of 250–500 nm, under high-dose electron radiation [157]. A pulsed electron beam with an energy of 130keV was employed at room temperature, with a pulse duration of 2ns and current density of $60\;/{\mathrm{cm}}^{2}$. The absorbed dose was 1.5kGy/pulse. Figure 8 shows the dose-dependent thermoluminescent (TL) intensity of the electron-irradiated MgO nanomaterials. Clearly, MgO’s structure should be modified by the electron radiation.

Unfortunately, there have been few studies on the irradiation effects of high-energy electrons on MgO-based MTJs. Metal layers are usually deposited over ferromagnetic layers of MTJs, preventing electrons from penetrating into MgO barriers and magnetic layers of MTJs. Therefore, high-energy electrons should not affect MTJs due to the screening effect of metal layers.

## 3. Effects of γ-ray Irradiation

γ-rays are a kind of electromagnetic radiation with wavelengths ranging from $3\times 10^{-13}\;m$ to $3\times 10^{-11}\;m$ (approximately 40keV to 4.0MeV), and being ionizing radiation. The electromagnetic wave can penetrate materials deeply and interact with matter through three kinds of primary processes: the photoelectric effect, Compton scattering, and electron–positron pair production, depending on the energy of the incident γ-ray. When the energy of the γ-ray is higher than 1.02MeV, it may spontaneously produce an electron and positron pair. Compton scattering is the principal mechanism when the energy of γ-ray is between 40keV and 4.0MeV. The photoelectric effect dominates when the energy of the γ-ray is below 50keV, whereby an electron absorbs the incident γ-ray and is excited to conduction bands. In all three kinds of processes, the γ-ray collides inelastically with electrons, losing energy and continuously moving with a longer wavelength. Furthermore, γ-rays can directly ionize atoms through the photoelectric effect and the Compton effect and indirectly through secondary ionization. These processes occur when MTJs are exposed to γ-rays.

Depending on the γ-ray’s energy and the properties of the MTJ materials, a γ-ray can induce displacements of atoms within the lattice, termed defects. These defects can remain for a long time at room temperature and can be investigated through the Hall effect and electrical measurements. This kind of radiation-produced defect would affect the performance of MTJs. In fact, most studies on MTJ degradation were performed under this kind of radiation interaction.

In contrast to the above interactions, γ-rays may only disturb the atoms of MTJ materials temporarily or transiently. The produced disturbances of atoms may disappear shortly once the γ-ray is removed. This kind of radiation-induced degradation can be only detected in-situ in real-time, while under irradiation.

Experimental investigations have supported these two kinds of γ-ray interactions. Several groups have reported that MgO-based MTJs are highly tolerant of γ-ray radiation up to a dose of 10Mrad [152,158]. In their work, MTJs were irradiated and then measured ex situ. Their results indicated that γ-ray irradiation did not noticeably change the TMR ratio, coercivity, or magnetostatic coupling of low-frequency noise. As such, MgO-based MTJ devices are expected to operate reliably in a γ-ray irradiation environment, especially at doses below a few hundred Rad [159]. Other scientists hold the view that γ-ray radiation should degrade MgO-based MTJ devices, because γ-rays have been shown to change the microstructure of MTJ materials [160]. Additionally, others suggested that γ-ray radiation may affect the peripheral circuits of MgO-based MTJ devices (not the MTJs themselves) during the read/write operation, leading to soft errors [161].

Table 7 lists some results of γ-ray irradiation. In the following subsections, these published data will be analyzed in detail with respect to the MTJ structures and experimental conditions, including the conditions of γ-ray irradiation and the measurement methods. First, the physical properties of the γ-ray-irradiated MTJ material, including MgO crystals (used as barriers in MTJs), fixed layers and free layers, and MgO/ferromagnetic layer interfaces, will be reviewed. Next, the review will focus on the physical properties of γ-ray-irradiated MgO-based MTJs. Finally, the tolerance ability of MgO-based MTJs will be discussed from γ-ray penetration in MTJs and MTJ devices, to explore potential explanations of MgO-based MTJs’ radiation tolerance.

### 3.1. MTJ Materials under γ-ray Irradiation

MgO-based MTJs consist of MgO barriers, ferromagnetic free/fixed layers, and metal electrodes. The performance of MTJs is influenced by their microstructures, physical properties, and the interfaces of these materials. Therefore, the characteristics of these MTJ materials with respect to γ-ray irradiation are discussed first.

#### 3.1.1. MgO Crystals under γ-ray Irradiation

There is a limited amount of literature available on the effects of γ-ray damage to MgO barriers with nanometer thickness [139]. To ensure adequate information on γ-ray-irradiated MgO materials, the properties of γ-ray-irradiated MgO bulk and thick films are reviewed here. It is expected that the properties of γ-ray-irradiated MgO barriers of nanometer thickness will exhibit similar behavior to that observed in MgO bulk and thick films.

The properties of irradiated MgO have been investigated since the 1960s to explore the potential of MgO for γ-ray dosimetry by studying its response to γ-ray irradiation. MgO crystals were cleaved from ingots and exposed to a ${}^{60}$Co source with a radiation intensity of $3.0\times 10^{6}\;R/h$ [162]. The thermoluminescence indicated γ-ray irradiation-induced defects in the MgO crystals.

MgO powder has also recently been irradiated by γ-rays. Kiesh et al. exposed commercial MgO powder to ${}^{60}$Co γ-ray radiation with a total dose of 20Mrads [164]. The irradiated powder was then kept at room temperature for over a year before measurements. The results showed that γ-ray irradiation caused a shift in the powder’s thermoluminescence peaks. In another study, MgO powder with a purity of 99.9% was exposed to γ-ray radiation, using a ${}^{60}$Co source with a dose rate of 8.33 mGy/s [165]. Figure 9a shows the thermoluminescence (TL) of the γ-ray-irradiated MgO powder. Low-dose γ-ray irradiation induced a peak around $280{\;}^{\circ }C$, while a high γ-ray radiation dose (above 300 Gy) resulted in a peak at $150{\;}^{\circ }C$, which became dominant after exposure to a dose above 1kGy. It was believed that the radiation dose affected the recombination centers and caused the shift in the TL peaks. Figure 9b shows the relationship between γ-ray dose and TL response integrated across the entire TL curve over the dose range. The TL response changed linearly with radiation dose at intermediate dose levels of 1–100 Gy, and sub-linearly at higher dose levels of 0.5–50 kGy.

Arshak et al. investigated MgO capacitors consisting of Ag electrodes and sandwiched MgO thick films [121,167]. The grain size of the MgO particles was 0.5–1.0 ${}^{\circ }$C. These MgO capacitors were exposed to γ-ray radiation with a maximum dose of 32.55mGy and an energy of 0.662MeV. Figure 10 shows the real-time capacitance of the MgO capacitors as a function of γ-ray radiation dose. The capacitance increased continuously with γ-ray dose, being reversible and less susceptible to γ-ray radiation. γ-rays damaged the MgO particles and produced structural defects (such as color centers or oxygen vacancies) in MgO, changing the density of charge carriers in the MgO films.

Steinike et al. exposed mechanically cleaved MgO samples to γ-rays emitted from a ${}^{60}$Co source [168]. The irradiation was carried out at a rate of 3.4–4.5 MRad/h and energy 1.25MeV at $-196{\;}^{\circ }C$. γ-ray irradiation generated F${}^{+}$ centers and V${}^{-}$ centers in the MgO crystals. The concentrations of F${}^{+}$ and V${}^{-}$ centers increased linearly with radiation doses up to 1–3 MRad, followed by saturation at higher doses. Additionally, the concentration of the F${}^{+}$ defect centers decreased with increasing annealing temperature and the F${}^{+}$ centers could be removed by annealing at $600{\;}^{\circ }C$.

Clement et al. studied the absorption and luminescence spectra of MgO crystals under γ-ray irradiation in-situ [169]. MgO crystals with 99.99% purity were exposed to γ-rays at a flux of $3.5\times 10^{4}\;\mathrm{rad}/h$ for 7h at $20{\;}^{\circ }C$ and $120{\;}^{\circ }C$, in a vacuum of less than $2\times 10^{-6}\;\mathrm{Torr}$. The real-time absorption increased with increasing radiation dose at both temperatures. It was also reported that subsequent annealing at $600{\;}^{\circ }C$ canceled the irradiation effect. Based on the results, it was concluded that impurities of Fe (less than 300 ppm) and Cr (less than 100 ppm) played a significant role in the degradation caused by irradiation.

Abramishvili et al. studied γ-ray-irradiated MgO crystals too [163]. The total impurity content in the crystals did not exceed 245ppm. The irradiation was carried out at room temperature, with a dose of $1.6\times 10^{6}\;\mathrm{rd}/h$ and a maximum γ-ray energy of 2.1MeV. in-situ measurements were performed at low temperatures. It was observed that the irradiation significantly changed the thermal conductivity of the MgO crystals, as shown in Figure 11a. The thermal conductivity was partially reversed by annealing the irradiated crystals at $515{\;}^{\circ }C$ for one hour, which led to the recovery of the crystals’ heat conductivity to their initial state. The observed reversal is consistent with other reports [169]. Additionally, their optical absorption was changed after irradiation, as shown in Figure 11b. Upon further analysis, it was believed that γ-ray radiation caused the formation of Frenkel pair defects, which changed both the thermal conductivity and optical absorption. Frenkel pair defects can be eliminated through annealing, which leads to the restoration of the original thermal conductivity.

Kvatchadze et al. measured, ex-situ, the thermo-stimulated luminescence (TSL) of nominally pure MgO single crystals containing minor impurities (Cr${}^{3+}$: 12–26 ppm; Mn${}^{2+}$: 35–72 ppm; V${}^{2+}$: 24–60 ppm; and OH${}^{-}$: 0–4.9 $\times 10^{17}$ cm${}^{-3}$) under γ-ray irradiation (0.8Gy/s and 1.25MeV) over a temperature range of 300K to 775K [160]. It was reported that in MgO crystals containing OH${}^{-}$ impurities, the TSL intensity steadily increased with increasing γ-ray radiation dose at 450K, as shown in Figure 12a. Additionally, the TSL intensity at 450K increased linearly with the γ-ray radiation dose (Figure 12b). However, in MgO crystals without OH${}^{-}$ impurities, the TSL intensity at 450K showed extremely low sensitivity to γ-ray irradiation (Figure 12b). It was proposed that foreign hydroxyl ions trapped charges in γ-ray-irradiated MgO crystals, inducing the accumulation of hole centers to change the optical properties.

Lynch et al. investigated the photoconductivities of MgO polycrystalline bulks under γ-ray irradiation fields in-situ, over a temperature range of 300K to 600K [170], as shown in Figure 13. The γ-rays were emitted from a ${}^{60}$Co source, with an energy of 1.17MeV or 1.33MeV. It was reported that the photoconductivity of the MgO bulks increased linearly with γ-ray radiation dose. The γ-ray-induced conductivity showed a linear dependence on the radiation dose rate up to $4.0\times 10^{5}\;\mathrm{rad}/h$. Additionally, the photoconductance of the MgO bulks increased by about three orders of magnitude when exposed to γ-ray radiation with a flux of (2.9–3.7) $\times \;10^{5}$ rad/h.

The studies described above demonstrate the sensitivity of MgO materials (single crystals, polycrystalline bulks, mechanically exfoliated layers, and powder) to γ-ray radiation. in-situ work indicated that radiation-induced defects could be restored to their initial pre-irradiated states, especially after being annealed at high temperatures.

The thickness of MgO crystalline films employed as barrier layers in MgO-based MTJ devices is only several nanometers. Such kinds of thin layers are expected to be more sensitive to γ-ray radiation than their bulk counterparts, as observed in other two-dimensional nanolayers [112]. As a result, MgO-based MTJs should be sensitive to γ-rays if MgO layers are exposed to γ-ray radiation.

Different from free-standing MgO films, MgO barrier layers are sandwiched between ferromagnetic free/fixed layers in MTJs. The ferromagnetic layers could potentially reduce the radiation dose into the MgO barrier layers and provide some level of protection. Furthermore, defects induced by γ-ray radiation may be temporary and disappear shortly after radiation exposure or thermal annealing. Therefore, the irradiation effects on MgO barrier layers are more complex than discussed above. in-situ and real-time measurements are required to examine the effect of γ-ray radiation on MgO barrier layers.

#### 3.1.2. Ferromagnetic Materials of MTJs under γ-ray Irradiation

Ferromagnetic films are utilized in MTJs to sandwich MgO barrier layers, with one being the fixed layer and the other being the free layer. These ferromagnetic layers are typically made of Fe(001) films, FeCo films, or CoFeB films. A typical MTJ consists of Si/SiO${}_{2}$/Ta(5)/Ru(10)/Ta(5)/Co${}_{20}$ Fe${}_{60}$B${}_{20}$(5)/ MgO(2.1)/Co${}_{20}$Fe${}_{60}$B${}_{20}$(3)/Ta(5)/Ru(5) (where the numbers in parentheses denote the thickness in nanometers) [171]. Unlike MgO barrier layers with a thickness of 1–2 nanometers, the ferromagnetic layers, such as CoFeB, are thick, and the performance of the MTJs is closely related to their magnetic properties.

Wang et al. [120] investigated CoFeB/MgO’s perpendicular-anisotropy magnetic tunnel junction and found that the magnetism was destroyed if the radiation dose was sufficiently high.

Shkapa et al. exposed FeCoB metallic ribbons to γ-ray radiation and examined their magnetic properties using nuclear magnetic resonance and the Móssbauer effect [118]. The (Co, Fe)${}_{85}$B${}_{15}$ metal glasses were irradiated by 1.2MeVγ-rays at $60{\;}^{\circ }C$. It was reported that Co${}_{85-x}$Fe${}_{x}$B${}_{15}$ (x=12−25) magnetic glasses were sensitive to γ-ray radiation, changing the atomic short-ordering of FeCoB ribbons.

Other ferromagnetic materials, such as Fe, Co, and FeCo alloys, should be similar to CoFeB materials under γ-ray irradiation. To ensure the similarity, the displacements per atom cross-section of Fe films with sizes of 100nm×3μm×12μm were calculated using the Monte Carlo simulation method [172] under γ-rays with energies of 1.3MeV and a source activity of 1000i. The displacement cross-section was 0.1barns. The calculation indicated that the atomic displacement rate was about 0.6/s. Furthermore, the γ-ray-induced displacement cross-sections were very low for γ-ray radiation with energy >1 MeV.

In addition to MgO barrier layers and ferromagnetic layers, non-magnetic metal films in MTJs, such as Ta and Ru layers, can protect MgO layers and ferromagnetic films from γ-ray radiation. However, the consequences of γ-ray irradiation of metal films are not the subject of this review. Although not discussed here, there is literature available on this topic [173].

#### 3.1.3. Interfaces of MgO Barrier/Ferromagnetic Layers

The performance of MTJs is influenced by interfaces between the MgO barriers and ferromagnetic layers. Recent investigations have shown that CoFeB can form Co(Fe)-O bonds and bond to MgO epitaxial grains after annealing [171]. Conversion electron Móssbauer spectroscopy studies indicated that interfaces between MgO(001) and Fe(001) layers were partially oxidized over 60%, and Fe diffused into MgO barriers from both ferromagnetic interfaces [174]. It has been suggested that these interfaces may be more sensitive to γ-ray radiation, similar to Al${}_{2}$O${}_{3}$-based MTJs, whose physical properties were significantly affected by irradiation [139].

Recent in-situ experiments discovered that the uniaxial magnetic anisotropy decreased systematically with increasing annealing temperature [175]. Specifically, the MgO/FeCoB/MgO layers become isotropic after annealing at $450{\;}^{\circ }C$. The asymmetry at the interfaces was explained by the diffusion of boron from the FeCoB interface layer into the adjacent MgO layer. The electronic structures of MgO/Fe interfaces have been investigated [176]. It is believed that Fe3d-O2p hybridization and distortion of the Fe film play important roles in magnetic anisotropy at the MgO/Fe interface.

Thermal annealing also affects the interfaces between MgO barriers and ferromagnetic layers. The details are discussed in the section on infrared radiation and thermal annealing.

### 3.2. MTJs under γ-ray Irradiation

Until this point in time, there have been two distinct viewpoints regarding the impact of γ-ray radiation on MgO-based MTJs. Some scientists believe that MgO-based MTJs are susceptible to γ-ray radiation and are likely to sustain damage as a result. Other scientists argue that MgO-based MTJs are resilient to γ-ray radiation. In the following subsections, each of these viewpoints will be reviewed in detail.

#### 3.2.1. Sensitivity Results

Considering the reported properties of MgO barrier materials and the discussion on the ferromagnetic layer materials of MgO-based MTJs above, it can be inferred that MgO-based MTJs would be affected by γ-ray radiation. However, there are limited reports on the degradation of MgO-based MTJs under γ-ray irradiation. Two sensitivity cases are reviewed below.

Wang et al. measured the magnetic properties of double-interface CoFeB/MgO perpendicular-anisotropy magnetic tunnel junctions (p-MTJ) [120]. The MTJ films were deposited on thermally oxidized Si substrates with CuN/Ta seed layers, consisting of [Co(0.5)/Pt(0.2)]${}_{\times 6}$/Co(0.6)/Ru(0.8)/Co(0.6)/[Co(0.5)/Pt(0.2)]${}_{\times 3}$/W(0.25)/CoFeB(1.0)/ MgO(0.8)/CoFeB(1.3)/W(0.2)/CoFeB(0.5)/MgO(0.75)/Ta(3.0)/Ru(8.0) (numbers in parentheses are thickness in nanometers). The CoFeB/MgO *p*-MTJs were exposed to a Cobalt-60 γ-ray radiation source at room temperature with a dose rate of 220rad/s. The results showed that the coercivity of the γ-ray-irradiated *p*-MTJs increased gradually with increasing doses of up to 20Mrad, as shown in Figure 14. However, there was no observed variation in the saturation magnetization.

It was reported that the magnetism of MgO-based MTJs was destroyed by γ-ray radiation when the dose was sufficiently high, such as 247Mrad [120]. It was hypothesized that the destruction of magnetism was caused by radiation-induced thermal stress. Figure 15 shows the surfaces of the MTJs after γ-ray irradiation, the observed effects were caused by differences in the thermal expansion coefficients between the MTJ films and the substrate.

#### 3.2.2. Tolerance Results

Numerous research groups have reported a high tolerance of MgO-based MTJs to γ-ray radiation, with no observed impacts on the magnetic or electrical properties of the MgO-based MTJs.

Nguyen et al. exposed a bare MTJ to γ-ray radiation (1.25MeV) for a total ionizing dose of 100kRad [166]. The MTJ consisted of Ru(7)/Ta(10)/Co${}_{60}$Fe${}_{20}$B${}_{20}$(3)/Mg(0.3)/MgO (1.1)/Co${}_{60}$Fe${}_{20}$B${}_{20}$(3)/Ru(0.8)/Co${}_{70}$Fe${}_{30}$(2.5)/PtMn (20)/Ta(5)/CuN(30)/Ta(5) (numbers in parentheses are thickness in nanometers). Ex-situ measurements revealed that the dose of γ-ray radiation did not cause any noticeable changes in the magnetic properties of the MTJ, as shown in Figure 16. The MTJ exhibited no noticeable changes in either coercivity or magnetostatic coupling.

Ren et al. investigated MgO-based MTJs exposed to γ-ray radiation [152]. The MTJs had a full structure of Si/Ru(6)/IrMn(11)/CoFeB(6)/MgO(1.4)/CoFeB(5) (numbers in parentheses are thickness in nanometers), as shown in the inset of Figure 17a. The tunnel barrier was made of MgO with (001) crystalline orientation. The junction was exposed to ${}^{60}$Co γ-ray radiation at a dose rate of 9.78rad/min. Figure 17a shows the hysteresis loop of a single MTJ before and after exposure to the γ-ray radiation. A 10MRad dose of radiation had a very weak effect on the electrical resistance. Figure 17b shows the coercive field H${}_{c}$ and TMR of other individual MTJs with the same structure that were tested under the same irradiation. The measured coercive field H${}_{c}$ and TMR were almost the same before and after γ-ray irradiation. Neither the electrical nor the magnetic properties of the MTJs were affected by the radiation. Therefore, the study concluded that MgO-based MTJs were highly tolerant of γ-ray radiation with a dose of 10MRad at 1.25MeV.

Hughes et al. exposed MgO-based MTJs to Co${}^{60}$γ-ray radiation with a dose of up to 1Mrad (Si) [119]. It was reported that γ-ray irradiation did not affect the state retention and switching characteristics of MgO-based MTJs.

Most of the experimental measurements discussed above were carried out after γ-ray radiation exposure, and it remains unclear whether the after-exposure status was equivalent to the exposure status. Nonetheless, it can be inferred that MgO-based MTJs are capable of retaining their non-irradiated initial status after γ-ray irradiation.

In addition to experimental studies, some theoretical research has been reported in support of the radiation tolerance of MgO-based MTJs. For instance, Kang et al. theoretically evaluated commercial CMOS nonvolatile units and MgO-based *p*-MTJs [161]. Their simulation results showed that CoFeB/MgO/CoFeB MTJs should be resistant to radiation effects.

### 3.3. Discussion of γ-ray Irradiation of MTJs

As mentioned above, certain research groups have claimed that MgO-based MTJs are sensitive to γ-ray irradiation in-situ due to the sensitivity of MgO barriers to γ-ray irradiation. On the other hand, most laboratories have reported that MgO-based MTJs are tolerant to γ-ray irradiation. In order to explain the discrepancy in the response of MgO-based MTJs to γ-ray irradiation, the effects of γ-rays are discussed below with respect to γ-ray penetration, the dynamic behavior of MTJ materials, and tunneling tolerance. The discrepancy may come from different experimental conditions.

#### 3.3.1. γ-ray Penetration in MTJs

MTJs consist of MgO barrier layers sandwiched between ferromagnetic free/fixed layers, and metal electrodes, as well as electrodes made from high atomic number (high Z) materials with high density, such as Ta and Au. Electromagnetic waves, including γ-rays, can pass through these metal and ferromagnetic layers to reach MgO barriers. In order to analyze this penetration under different irradiation conditions, electromagnetic penetration is calculated. The intensity of electromagnetic radiation inside MTJs decreases exponentially from the MTJ’s surface, as described by the equation based on the Beer–Lambert law [177,178]:(4)$$I=I_{0}e^{-\mu z}$$
where *I* is the intensity of electromagnetic radiation transmitted over a distance *z*, $I_{0}$ is the incident electromagnetic wave intensity, μ is the linear attenuation coefficient in cm${}^{-1}$, $\mu =n\sigma =n(\sigma_{photoelectric}+\sigma_{Compton}+\sigma_{Pair})$ (*n*: the number of atoms/cm${}^{3}$; σ: proportionality constant that reflects the probability of an electromagnetic wave photon being scattered or absorbed), and *z* is the distance traveled by the radiation in cm. For multilayered films, the electromagnetic intensity is proportional to both the attenuation coefficient and the thickness of each layer through which it passes [179].

The calculation of electromagnetic radiation transmission through an MTJ is based on a typical MTJ structure consisting of Ta(5)/Ru(10)/Ta(5)/Co${}_{20}$Fe${}_{60}$B${}_{20}$(5)/MgO(2)/Co${}_{20}$ Fe${}_{60}$B${}_{20}$(3)/Ta(5)/Ru(5)/Cr(10,000)/Au(10,000) (with numbers indicating nominal thicknesses in nanometers), as described in the published literature [171,180]. The linear attenuation coefficients of each film material are obtained from published data and used in the calculation. Equation (Equation 4) is then applied to calculate the transmission of electromagnetic radiation through the MTJ device. Figure 18 shows the calculated electromagnetic radiation intensity in a typical MTJ structure. The used electromagnetic radiation spans from 4.950keV to 1MeV in energy, covering both γ-ray (with energy greater than 124keV) and X-ray (with energies of 125 eV–125 keV) radiation. According to the theoretical calculations, γ-rays could penetrate the entire MTJ structure without undergoing significant absorption.

Some MTJs may contain thick metal electrodes, which can affect the penetration of γ-rays through the devices. Figure 19 shows the transmission of γ-rays through iron, a ferromagnetic material used in some MgO-based MTJs [2,174,181]. γ-rays can penetrate through iron for several centimeters, consistent with other reports [182]. Thus, γ-rays with various energy levels can easily penetrate entire MTJs, which consist of metal nano-films and thick electrodes, after passing through the top electrodes. This suggests that MTJs can be penetrated by γ-rays and their metal layers cannot shield all γ-rays, especially those with high energies.

#### 3.3.2. Possible Explanations of Radiation Degradation

Based on the discussion of the radiation penetration above, it can be concluded that when exposed to γ-rays, MgO barriers should undergo interactions with γ-rays, i.e., photoelectric effects, Compton scattering, and electron–positron pair production, as discussed in previous sections. These interactions would cause displacement of Mg atoms or O atoms within lattices, resulting in defects or amorphizations of MgO barriers. As a consequence, MTJs would experience γ-ray-induced degradation according to the Julliére model.

Figure 20a illustrates the Julliére coherent tunneling in an MTJ with a crystalline MgO barrier and two ferromagnetic layers. The tunneling process involves three kinds of Bloch states with different wave function symmetries existing in the free/fixed layers, which pass through the MgO barrier. The high MR ratio of the Fe/MgO/Fe sandwich structure primarily depends on the coherent spin-dependent tunneling that occurs in the crystalline MgO(001) tunnel barrier.

Irradiation can have an impact on tunneling. Figure 20b demonstrates the tunneling through an amorphous barrier. When the MgO(001) tunnel barrier becomes amorphous due to irradiation, the crystallographic symmetry of the tunnel barrier is lost; Bloch states with various symmetries can couple with the MgO tunneling states, resulting in finite tunneling probabilities. In 3d ferromagnetic metals and alloys, Bloch states with $\Delta_{1}$ symmetry (*s*-*p*-*d* hybridized states) generally exhibit a large positive spin polarization *P* at the Fermi energy $E_{F}$, while those with $\Delta_{2}$ symmetry (*d* states) tend to have a negative spin polarization *P* at $E_{F}$ [20,183]. All Bloch states in the ferromagnetic free/fixed layers contribute to the tunneling current, affecting the net spin polarization of the ferromagnetic layers and degrading the functionalities of MTJ devices. In other words, after γ-ray irradiation, the momentum of tunneling electrons is no longer conserved due to local disorder scattering. This would destroy the coherence or symmetry of conducting electrons and changes the coherent tunneling process to incoherent tunneling through the displacement of atoms, degrading MTJs. It was experimentally proved that defects in MgO barriers impacted polarized tunneling, localized states of spin, and polarized symmetry tunneling across MgO barriers [184]. The electronic properties of MgO grain boundaries in MTJs are symmetry dependent [185].

In addition, the energy of γ-rays can be transferred to electrons, resulting in an increase in the number of high-energy free spin electrons that interact with the lattices and interfaces. This increase can change the spin polarization:(5)$$P=\frac{N_{↑}\left(E_{F}\right)-N_{↓}\left(E_{F}\right)}{N_{↑}\left(E_{F}\right)+N_{↓}\left(E_{F}\right)}$$
here, $N_{↑}\left(E_{F}\right)$ and $N_{↓}\left(E_{F}\right)$ are the density of state at the Fermi energy ($E_{F}$) for spin-up electrons and spin-down electrons, respectively. γ-rays can change the density of states at $E_{F}$, affecting spin- and polarized-symmetry tunneling. Additionally, γ-rays can penetrate through the free/fixed layers, modifying their electrical and magnetic properties through the photoelectric effect and the Compton effect, as well as by indirect ionization, which can intermittently or permanently degrade MTJ performance.

According to the Julliére model and the penetration analysis, MgO-based MTJs are expected to degrade under γ-ray irradiation, as reported in some literature. Briefly, MTJs should be sensitive to γ-ray radiation.

#### 3.3.3. Possible Explanations of Tunneling Tolerance

Most research groups have reported that MgO-based MTJs are highly tolerant to γ-ray radiation and not degraded by γ-ray radiation at all. There are three possible reasons for this.

One possible explanation of their tolerance to γ-rays is that the unique TMR mechanism of MTJs enables MgO-based MTJs to be tolerant. The tunneling mechanism is the most popular explanation. The magnetic properties of MTJs originate from spin charges, which makes MTJs resistant to radiation. While γ-ray radiation can amorphize MgO barriers and ferromagnetic fixed/free layers, the resulting partial amorphous status of MTJ layers has only a *slight* effect on the magnetic characteristics of the fixed/free layers [171]. Therefore, the degradation caused by radiation in MgO-based MTJs is negligible.

Secondly, the degradation of MgO-based MTJs is limited to specific conditions, such as exposure to extremely high doses of radiation, which can result in complete or partial destruction of the crystallographic structures of the MTJ layers and cause MTJs to loss their functionality. Fortunately, such critical conditions are rare in γ-ray irradiation, although they can occur in neutron irradiation and high-energy ion irradiation. Therefore, the degradation of MTJs under γ-ray irradiation is expected to be minimal and maybe not to be detected.

Thirdly, many reported measurements have been carried out ex-situ. As discussed below, the damage caused by irradiation may diminish over time, and the physical properties of MTJs may be restored by the time measurements are taken.

#### 3.3.4. Possible Explanations for Divergence

It is evident that γ-rays can modify MgO barriers and ferromagnetic layers, and some research groups have reported degradation of MgO-based MTJs as a result. Additionally, the fact that most MTJs cannot operate at high temperatures suggests that MTJs are susceptible to infrared electromagnetic waves, which have lower energy than γ-rays. The temperature-induced degradation indirectly indicates that MTJs should be susceptible to γ-rays, with higher energy. However, most research reports have indicated that MgO-based MTJs are tolerant to γ-rays. The divergence among these reports may be explained in the dynamic properties of γ-ray-induced damage, which can account for the divergence among the literature.


*Temporary Defects*


Like high-energy ion radiation, γ-rays may only induce temporary defects that do not persist for a long time at room temperature and vanish after exposure to radiation. The excited electrons and ionization can quickly return to the initial state due to thermal motion at room temperature. Figure 13 shows one case where γ-rays changed a physical property (photoconductivity) of MgO, which was restored to its initial state after γ-ray irradiation due to thermal motion.

Figure 21 shows another case. The TL intensity of MgO powder irradiated by γ-rays was measured immediately or 75 days after irradiation [165]. The signals induced by the γ-ray radiation diminished over time. Recent calculations have also demonstrated that thermal motion at room temperature can eliminate the impact of γ-ray irradiation.

Due to the time-dependent dynamic nature of the impacts of γ-rays, only in-situ measurements can detect the transiently degraded performance of MTJs with excited states. Some studies have reported soft errors of MgO-based MTJs in in-situ measurements under irradiation [161], which were consistent with the assumption of temporary impacts over time.

Until now, most state-of-the-art measurements have been carried out ex-situ, without the presence of γ-rays. Some measurements were performed immediately after γ-ray irradiation, such as within 2min after removal of the radiation sources [162], or long after irradiation, such as after one year of storage at room temperature [164]. The impact of the irradiation may have diminished prior to making the measurements. Impact information under γ-rays may decay over time and become undetectable. This may be one explanation for why MgO-based MTJs have been reported to be resistant to γ-ray radiation in certain instances.

γ-ray irradiation may only transiently change the physical properties of MgO-based MTJ layers during irradiation procedures and not cause permanent damage. The properties of MTJ layers can be restored *reversibly* after exposure to γ-ray radiation, and therefore, MTJs can return to their initial state before irradiation. The temporary degradation of MgO-based MTJs induced by γ-ray radiation is not detectable in ex-situ measurements.


*Irradiation Annealing*


Irradiation annealing may eliminate the impacts of irradiation. High-energy γ-ray radiation can produce permanent defects in MgO barriers and ferromagnetic layers, changing their crystallographic structures and the physical properties of the layers, thereby degrading the performances of γ-ray-irradiated MTJs. However, these defects may revert to their initial equilibrium state over time at high temperatures. High-dose-rate γ-ray radiation can generate such high temperatures in MgO-based MTJs. The radiation-induced heat can self-anneal MTJs, erasing the effects of γ-ray irradiation and preventing the degradation of γ-ray-irradiated MTJs.

Regrettably, there are few experimental reports on irradiation annealing. The temperature of MgO-barriers and free/fixed layers is rarely mentioned in the literature, and the time interval between γ-ray irradiation and physical measurements is also unknown. More comprehensive in-situ and real-time investigations on the interactions between γ-rays and materials are required.

## 4. Effects of Lower-Energy Irradiation

Electromagnetic waves with wavelengths longer than gamma rays are commonly known as lower-energy waves, such as X-rays, ultraviolet radiation (UV), visible light, infrared radiation, microwaves, and radio waves. These electromagnetic waves have less energy compared to gamma rays, and are generally classified as non-ionizing radiation, with the exception of X-rays.

### 4.1. X-ray Irradiation

The energy of X-rays ranges from several tens of electron volts to hundreds of kiloelectron volts. The intensity of X-rays decreases exponentially from the surfaces of MTJs, as described by the Beer–Lambert law in Equation (Equation 4). X-ray radiation typically only penetrates a few microns into materials, depending on its energy and the material’s composition. MgO-based MTJs are typically sandwiched by electric electrodes made of materials such as gold or tantalum. These metal electrodes are usually thick enough to prevent X-rays from penetrating through to the MgO barriers and ferromagnetic layers of the MTJs. The detailed screening effect can be calculated. Figure 18 shows the calculated penetration intensity of X-ray radiation with an energy above 4.950keV (energy of X-ray: 124.8 eV–124.8 keV). Hard X-rays can fully penetrate MgO-based MTJs with weak absorption, therefore affecting the physical and chemical properties of both the MgO barriers as well as the ferromagnetic layers. The MgO barrier layers should be affected by X-ray radiation in a similar way to two-dimensional MoS${}_{2}$ monolayers [112,140]. In this case, the effects of X-ray radiation on MgO-based MTJs are very similar to those of γ-ray radiation. These X-ray effects may also be temporary and only detectable through real-time measurements. Soft X-rays with energies of ten kiloelectron volts or less would be strongly screened by metal electrodes, with penetration through to the MgO barriers and ferromagnetic layers of MTJs being prevented. Consequently, the effects of soft X-ray irradiation can be disregarded. Up to now, there have been few studies of X-ray radiation on MgO-based MTJs.

### 4.2. UV–Vis Irradiation

The energy of ultraviolet–visible (UV–vis) electromagnetic waves ranges from 1 eV to several tens of electron volts, with wavelengths of 10–400 nm. As shown in Figure 18, UV and visible electromagnetic waves cannot penetrate through metal layers to reach ferromagnetic and MgO layers. Additionally, metallic electrodes reflect UV–vis radiation, making MgO-based MTJs highly resistant to such radiation.

However, heat produced by UV–vis radiation may degrade MgO-based MTJs. Doped MgO materials have been studied as a potential material for UV dosimetry to detect ultraviolet radiation [186,187]. Figure 22 shows the thermoluminescent (TL) response of UV-irradiated MgO crystals. Studies have shown that the thermoluminescent peaks of doped MgO crystals depend significantly on the dose of ultraviolet radiation with wavelengths such as 295nm [186], 289nm [187], and 249nm [187]. Even pure MgO crystals are affected by ultraviolet radiation with wavelengths such as 295nm[186] and 337nm [188]. Similar behavior was reported at other ultraviolet wavelengths [187]. These studies demonstrate that UV radiation changes the microstructure of MgO materials. However, the specific physical processes underlying the UV–vis radiation and MgO materials have not been not well described in the literature. The most likely explanation, is that UV–vis radiation causes an increase in temperature in the MgO materials, leading to their degradation.

It is noteworthy that the changes in the TL signals induced by UV–vis irradiation decreased over time. It was reported that the TL intensity of some irradiated crystals restored up to 95% of its initial value after being stored at room temperature for four days [187].

In theory, UV–vis radiation should degrade MgO-based MTJs, because MgO is sensitive to these electromagnetic waves. However, this degradation should only be temporary and result from radiation-induced heating. If heating effects are avoided, MgO-based MTJs should be highly tolerant to UV–vis radiation. To date, there is no literature available on the subject of the effects of UV–vis radiation on MgO-based MTJs.

### 4.3. Infrared Radiation and Thermal Annealing

Heat radiation or thermal radiation is a well-known term for infrared radiation. Pulsed thermal radiation, with a long wavelength of 1–20 microns and energy of 1–24 eV, can be efficiently screened by metallic electrodes. However, continuous thermal radiation, also known as heat, can penetrate MTJ devices during prolonged exposure to high temperatures, resulting in thermal annealing and thermal equilibrium. Thus, infrared radiation is somewhat different to other types of radiation.

There are reports on the annealing effect on MTJ component materials. Nikiforov et al. studied the pulse cathodoluminescence (PCL) excitation of MgO nanomaterials with a size of 250–500 nm [157]. It was reported that the PCL intensity in the 2.0–3.5 eV band increased by an order of magnitude with increased annealing temperatures, attributed to the relaxation of F-type centers (oxygen vacancies with two captured electrons). Shen et al. investigated the impact of thermal annealing on ferromagnetic CoFeB layers [189]. Their investigation indicated that thermal annealing enhanced the crystallization of CoFeB at the interfaces with MgO, affecting the magnetoresistance of MgO-based MTJs. Yuasa et al. reviewed the annealing effect on CoFeB electrodes [31], and interested readers are referred to the literature cited therein.

Ikeda et al. investigated the effect of thermal annealing on MTJs at temperatures higher than $500{\;}^{\circ }C$ [30]. The MTJs have a structure of Ta(5)/Ru(10)/Ta(5)/Co${}_{20}$Fe${}_{60}$B${}_{20}$ (5)/MgO(2.1)/Co${}_{20}$Fe${}_{60}$B${}_{20}$(4)/Ta(5)/Ru(5) (in nm). It was reported that the annealing process led to the relaxation of residual stress and an improvement in the (001) orientation of the MgO barriers, resulting in an enhanced TMR ratio.

Wang et al. studied both in-situ and ex-situ measured TMR values at $380{\;}^{\circ }C$[190]. The TMR structure consisted of Si/SiO${}_{2}$/Ta(7)/Ru(20)/Ta(7)/CoFe(2)/IrMn(15)/CoFe(2) /Ru(1.7)/CoFeB(3)/MgO(1.5–3)/CoFeB(3)/Ta(8)/Ru(10), with the numbers indicating the layer thicknesses in nanometers. It was found that the amorphous CoFeB layers underwent crystallization, and the quality of the MgO barriers’ crystallinity improved in less than 10min of annealing, resulting in a TMR value larger than 200%. The crystallization was further experimentally confirmed through their HRTEM work [171].

Liu et al. investigated the thermal stability of MTJs with MgO barriers at temperatures up to $500{\;}^{\circ }C$ [191]. The MTJs consisted of Ta(30)/[Co${}_{50}$Fe${}_{50}$]${}_{\times 3}$/IrMn(15)/[Co${}_{50}$Fe${}_{50}$]${}_{\times 2}$/ Ru(0.8)/[Co${}_{40}$Fe${}_{40}$B${}_{20}$]${}_{\times 3}$/MgO(1.2)/[Co${}_{40}$Fe${}_{40}$B${}_{20}$]${}_{\times 3}$/Ta(10)/Ru(5). The study observed the irreversible loss of magnetoresistance at high temperatures.

Typically, thermal annealing (using infrared radiation) has a positive effect on the crystallization of MgO barriers, which enhances the performance of MTJs. However, thermal annealing also accelerates interface diffusion between MgO barriers and ferromagnetic layers, leading to degradation of MTJ performance [192]. Xu et al. employed transmission electron microscopy and electron energy loss spectroscopy to investigate the microstructures of the MgO-CoFeB interfaces of MTJs [193]. Figure 23 shows HRTEM images, STEM images, and EELS mapping of the interfaces after thermal annealing. Thermal annealing indeed crystallized MTJ layers, as shown by the HRTEM images, and caused boron diffusion. Boron diffusion led to the growth of CoFe nanocrystals from CoFeB layers under annealing, while the crystallization did not significantly affect the MR properties. Instead, the MR ratio was predominantly determined by grain boundary transport caused by boron distribution. If boron diffused to metallic underlayers from the inside to the outside (as shown in Figure 23e,f), the MR ratio would be improved. Conversely, annealing may result in boron diffusing into grain boundaries of the MgO barriers from the outside to the inside (shown in Figure 23g,h), leading to a decrease in the MR ratio. The interfacial properties of MTJs regulated the diffusion of boron and affected the effect of thermal annealing. Thus, the effect of thermal radiation on MTJ devices depends on the annealing temperature, the duration, and the structure of the MTJs. Thermal irradiation can either benefit or degrade MTJs’ performance.

It is important to note that radiation other than infrared radiation can also produce heat, particularly at high-dose rates, which can lead to an increase in the temperatures of MTJs and produce similar annealing effects. Under such circumstances, high-energy radiation, such as γ-ray and hard X-ray radiation, may cause additional annealing effects. To study the effects of irradiation, it is crucial to investigate MTJs at constant temperatures or monitor the internal temperatures of MTJs, particularly the temperatures of the MgO and ferromagnetic layers.

### 4.4. Microwave Irradiation

The penetration depth of microwaves into conductive metal surfaces is typically less than one micron [194]. Therefore, the metallic electrodes of MTJs can efficiently reflect microwaves. In other words, microwaves should not penetrate through the electrodes to irradiate the MgO barriers and ferromagnetic layers. Therefore, the microwave irradiation effect can ignored, and microwave radiation should not have any significant impact on the performance of MTJs.

Although microwave radiation is not expected to penetrate through the electrodes of MTJs to affect the MgO barriers and ferromagnetic layers, it can cause a significant increase in the temperature of metal layers. Research has shown that microwave irradiation can produce a high temperature, of up to $500{\;}^{\circ }C$, in Au films in less than 10 s [195]. Therefore, microwave irradiation can generate a high temperature locally in ferromagnetic fixed-/free-layes of MTJs, which can have a significant impact on the performance of MTJs.

Up to now, there have been limited reports on the impact of microwave radiation on MgO-based MTJs. Some groups have investigated the behavior of MgO-based MTJs under microwave irradiation [94]. Unfortunately, it was not stated whether the MgO-based MTJs were damaged under microwave irradiation.

### 4.5. Radiofrequency Electromagnetic Irradiation

Radiofrequency (RF) electromagnetic radiation can be shielded by conductive or magnetic materials, which is known as RF shielding. Since MTJs have metal electrodes, these electrodes can block RF radiation and therefore MTJs should not be affected. The theoretical calculation shown in Figure 18 also predicts that electromagnetic waves with energy lower than four kiloelectron volts would not penetrate through the electrodes of MTJs. As listed in Table 1, the energy of radiofrequency radiation is typically less than a few milli-electronvolts, so RF radiation should be totally shielded and not affect MTJ performance.

Similar to microwaves, RF irradiation can also induce heating in metals, leading to high temperatures locally in MTJ electrodes. However, the induced temperature is expected to be low due to the extremely low energy of RF radiation.

Therefore, the effects of radiofrequency and other electromagnetic irradiation with longer wavelengths can be ignored. MgO-based MTJs should be highly tolerant to this radiation.

## 5. Outlook

MgO-based MTJs are promising for various applications, such as MRAM in quantum computers, logic gates, ultra-sensitive sensors, and energy harvesting and storage. These devices can be utilized in space technology, and therefore, the impact of radiation is crucial. With advancements in super-large-scale integration (SLSI) technology for central processing units (CPUs) and graphics processing units (GPUs) and programming languages such as the open source Python programming language, as well as professional packages/libraries for programming languages, it is possible to simulate complex interactions between radiation and MTJ components at the atomic level. Dynamic simulations at the atomic level can be employed to investigate individual atomic motion and nanoscale displacement under irradiation, and to calculate MR, providing insights into the dynamic behavior of atoms during irradiation. Additionally, the development of artificial intelligence (AI), including machine learning and deep learning, makes it possible to collect most research data on the irradiation of MgO-based MTJs and to systematically analyze the radiation’s impacts. Various parameters, such as radiation energy, duration, dose, and dose rate, can be simulated, investigated, and compared with experimental data to understand the electromagnetic–material interaction. Safe operation of MgO-based MTJs can be predicted in various irradiation environments.

## 6. Conclusions

The effects of radiation on MgO-based magnetic tunnel junctions have been reviewed and analyzed in various irradiation environments, including high-energy cosmic radiation, gamma-ray, X-ray, UV–vis, infrared, microwave, radiofrequency, and long-wavelength electromagnetic radiation. The examination considered both the material properties and device performance. In general, cosmic radiation (including ions and protons) can damage MTJs due to permanent atom displacements in the MTJ layers. While some groups have reported that γ-ray irradiation degrades the performance of MgO-based MTJs, the majority of scientists have claimed that MgO-based MTJs are tolerant to γ-rays without significant degradation in their performance. The impact of hard X-ray irradiation is comparable to that of γ-ray irradiation. Soft X-ray, UV–vis, infrared, and microwave radiation can be screened or shielded by the metal electrodes of MTJs, and these types of electromagnetic radiation should not significantly affect MTJ devices. Nonetheless, these types of radiation may induce heat or annealing, especially for infrared and microwave radiation, which can affect MTJ performance by causing crystallization of the MgO barriers and ferromagnetic layers as well as interfacial diffusion. There is no strong evidence that the present MgO-based MTJ devices are susceptible to radiation. The effects of radiation on MgO-based MTJs are discussed with respect to electromagnetic penetration, the Julliére model, the TMR mechanism, and annealing, to explore the physics behind these reported experimental data. Further in-situ and real-time investigations are necessary to fully understand the radiation tolerance of MgO-based MTJ devices under various types of electromagnetic radiation.

## Figures and Tables

**Figure 1 molecules-28-04151-f001:**
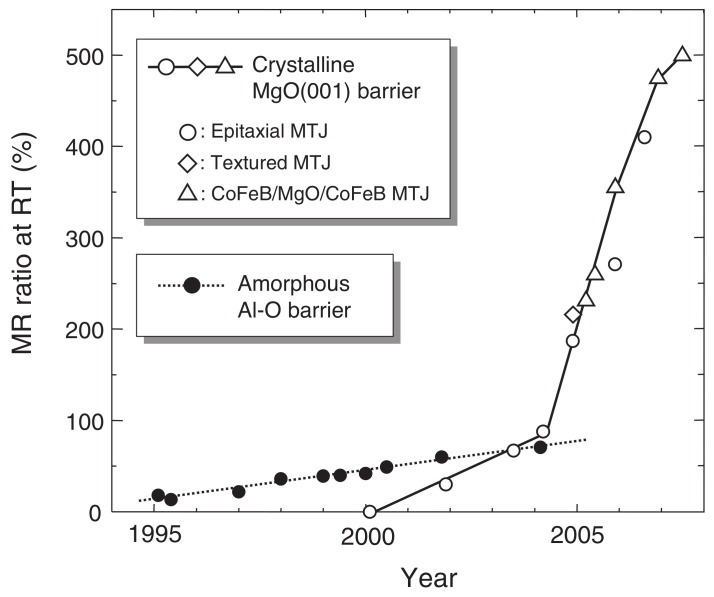
Historical development of MR ratio of MgO-based MTJs at room temperature. The data of AlO-based MTJs are also plotted for comparison. Reproduced with permission [15]. Copyright 2008, the Physical Society of Japan.

**Figure 2 molecules-28-04151-f002:**
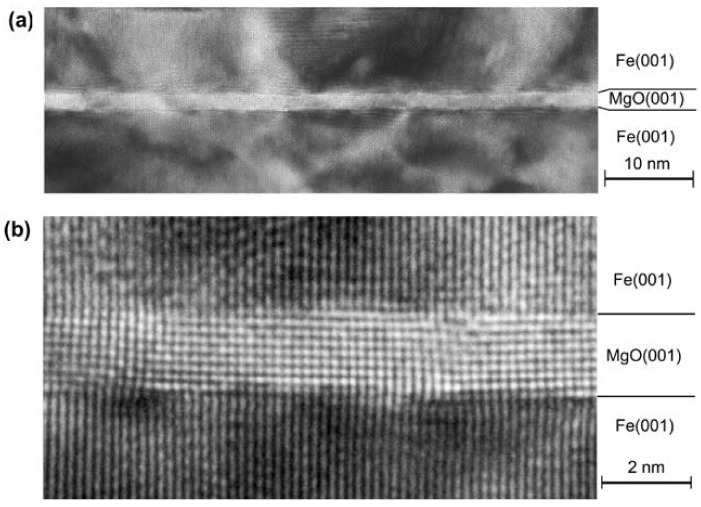
(**a**) TEM and (**b**) HRTEM images of an Fe(001)/MgO(001)/Fe(001) MTJ. Reproduced [2]. Copyright 2004, Springer Nature.

**Figure 3 molecules-28-04151-f003:**
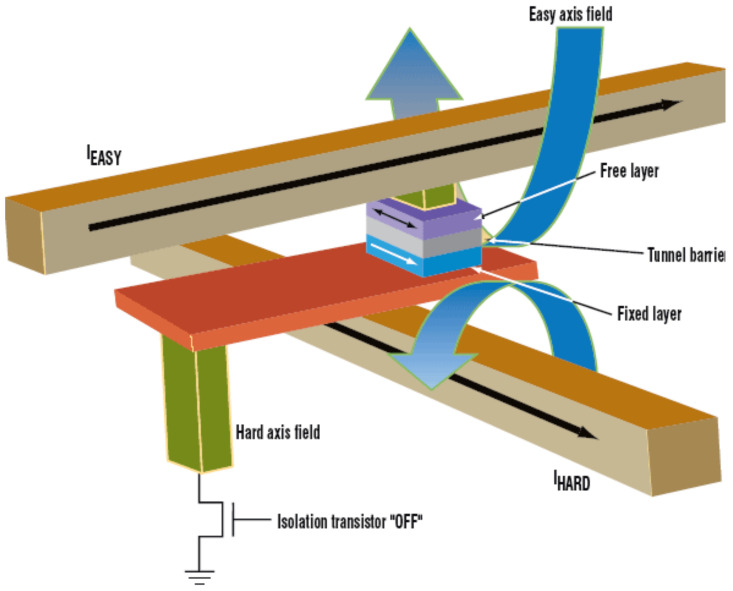
Structure of an MRAM cell (courtesy of Freescale).

**Figure 4 molecules-28-04151-f004:**
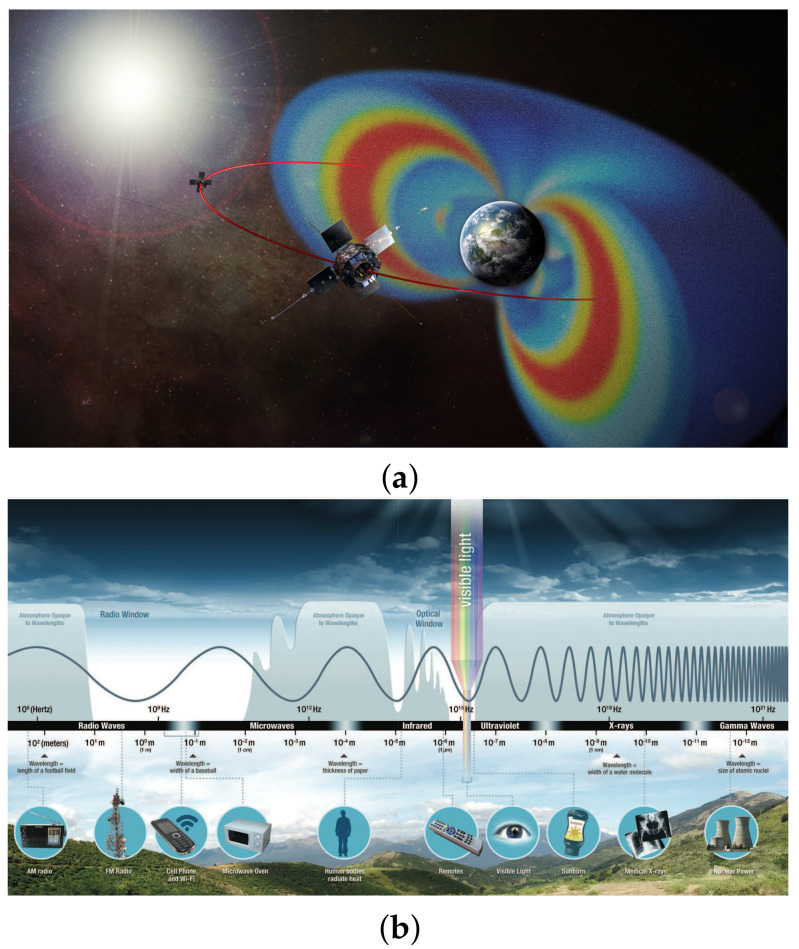
Radiation (**a**) in outer space and (**b**) on Earth. Satellites are orbiting in the radiation zone of the Van Allen belts whose cross-sectional shape and intensity are shown in (**a**). From nasa.gov (accessed on 7 August 2017).

**Figure 5 molecules-28-04151-f005:**
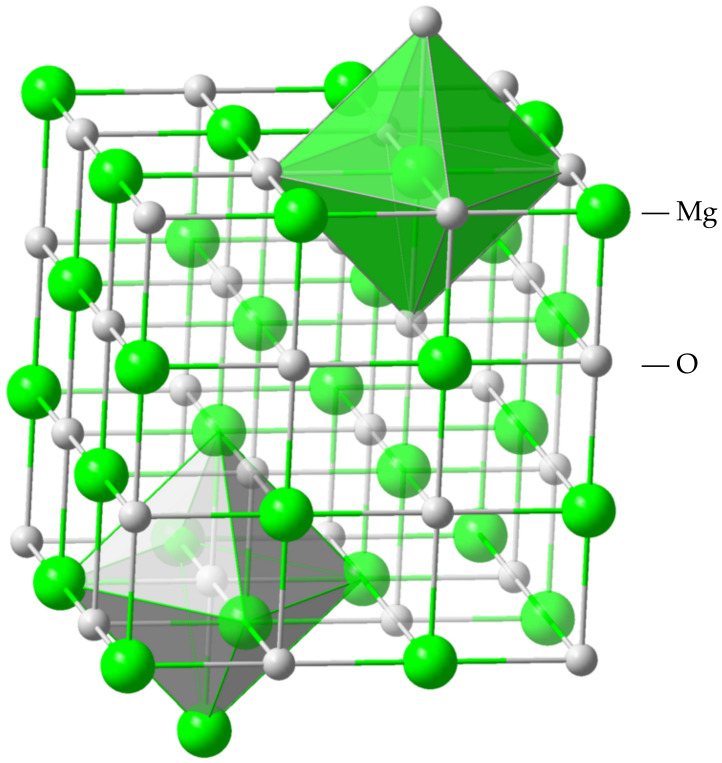
Crystallographic structure of MgO.

**Figure 6 molecules-28-04151-f006:**
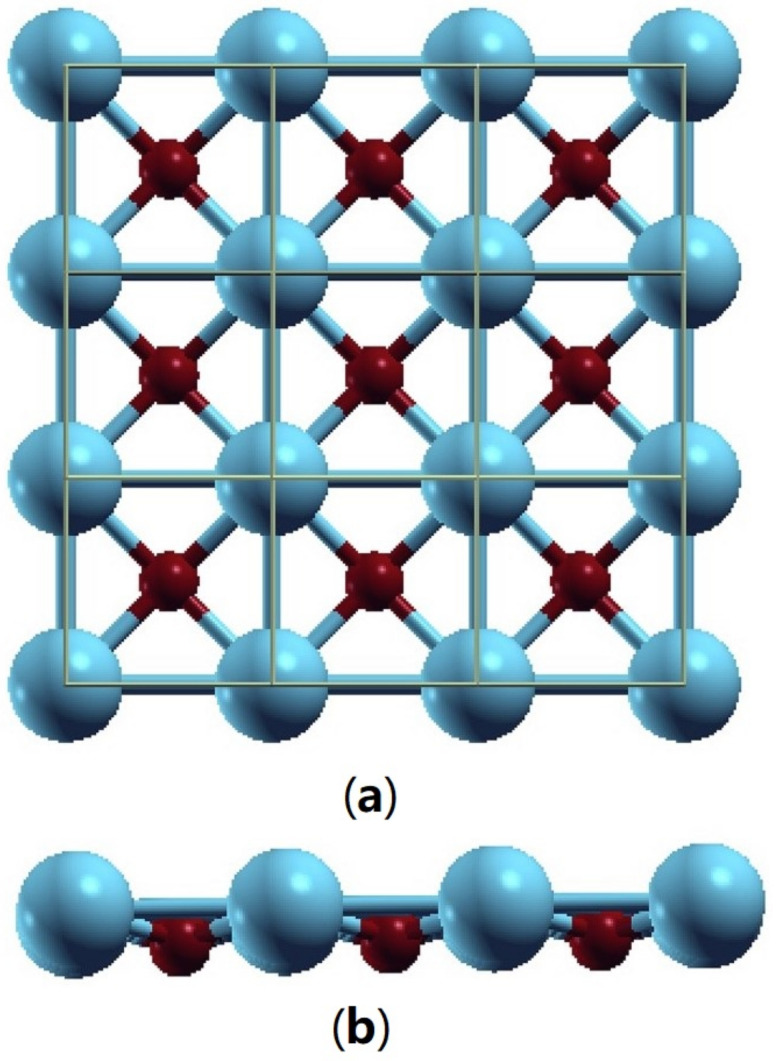
(**a**) Top view (along <001> direction) and (**b**) side view (along <100> direction) of a MgO (001) monolayer.

**Figure 7 molecules-28-04151-f007:**
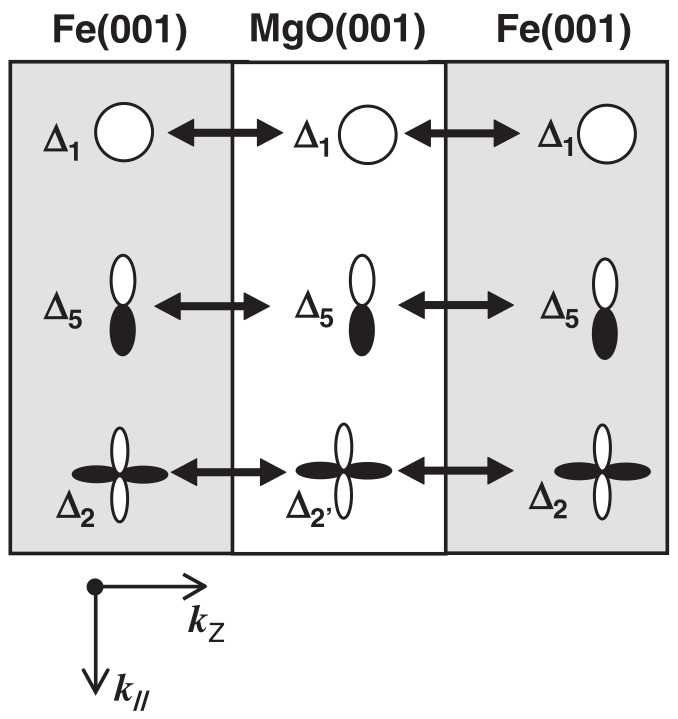
Coupling of wave functions between the Bloch states in ferromagnetic Fe(001) layers and the evanescent states in the MgO(001) barrier for $k_{\|}=0$ direction. $\Delta_{1}:s-p-d$; $\Delta_{2}:d$; $\Delta_{5}:p-d$. Reproduced with permission [15]. Copyright 2008, the Physical Society of Japan.

**Figure 8 molecules-28-04151-f008:**
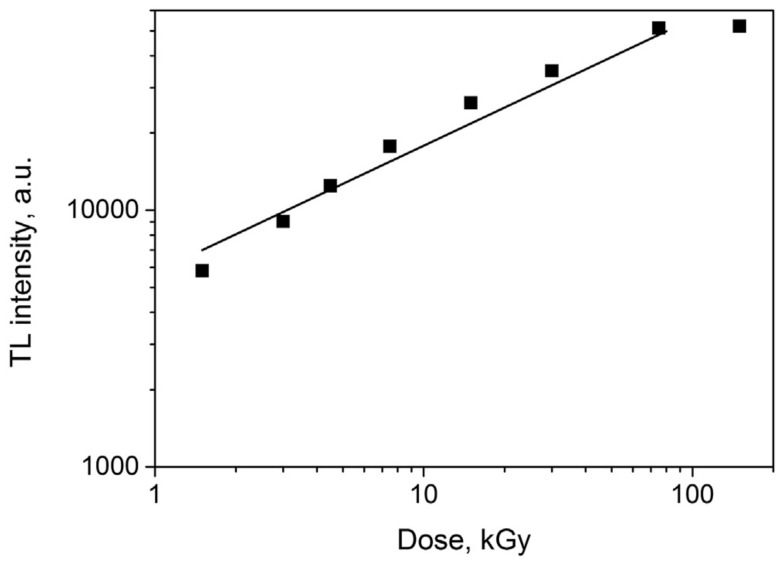
Dose dependence of TL intensity of MgO nanomaterials irradiated by a pulsed electron beam. Reproduced with permission [157]. Copyright 2015, Elsevier Ltd.

**Figure 9 molecules-28-04151-f009:**
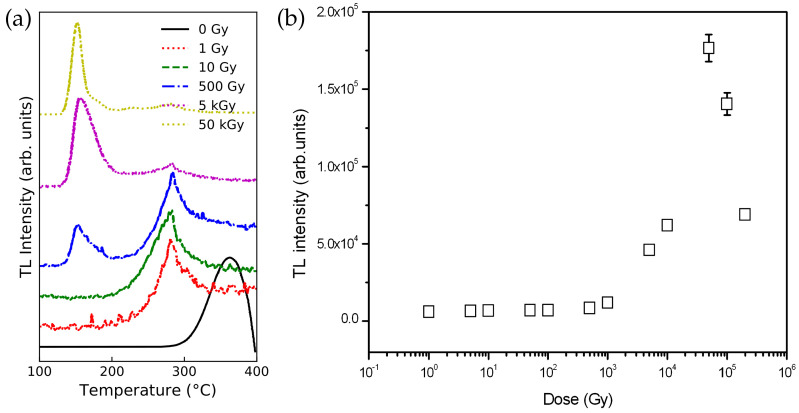
(**a**) TL intensity of MgO powder irradiated by gamma rays. Replotted from [165]. (**b**) TL response of MgO powder with gamma-ray dose. Reproduced with permission [165]. Copyright 2009, Taylor & Francis Group.

**Figure 10 molecules-28-04151-f010:**
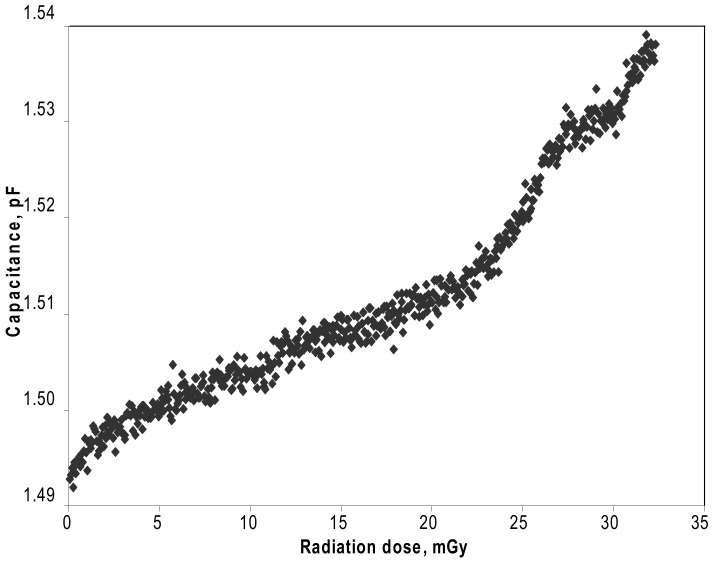
Real-time capacitance versus γ-ray radiation dose for Ag/MgO/Ag capacitors. Replotted from Ref. [121]. Copyright 2005, Springer.

**Figure 11 molecules-28-04151-f011:**
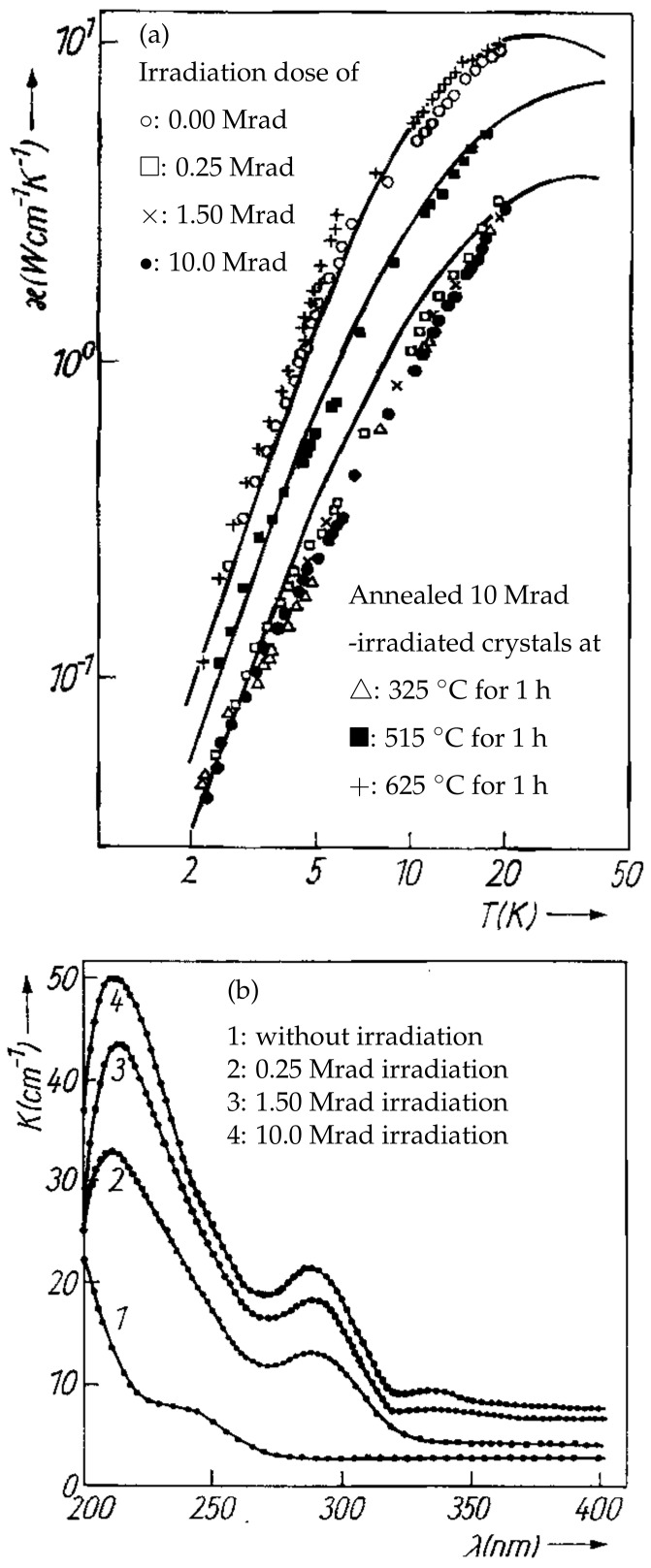
(**a**) Thermal conductivity and (**b**) spectra of optical absorption of MgO crystals before and after γ-ray irradiation. Reproduced with permission [163]. Copyright 1981, John Wiley and Sons.

**Figure 12 molecules-28-04151-f012:**
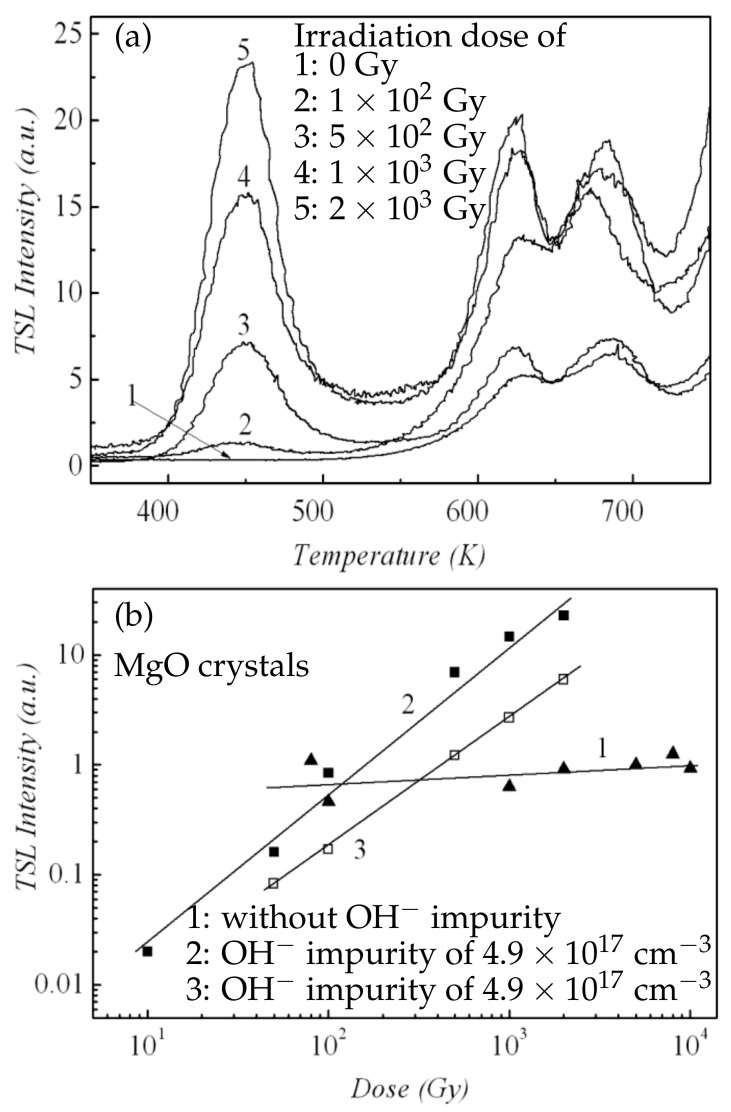
(**a**) TSL curves of MgO single crystals with OH${}^{-}$ impurity of $4.9\times 10^{17}\;/{\mathrm{cm}}^{3}$ under γ-ray irradiation under different temperatures. (**b**) TSL intensity dependence of γ-ray irradiation dose at 450K [160]. Copyright 2011, David Publishing Company.

**Figure 13 molecules-28-04151-f013:**
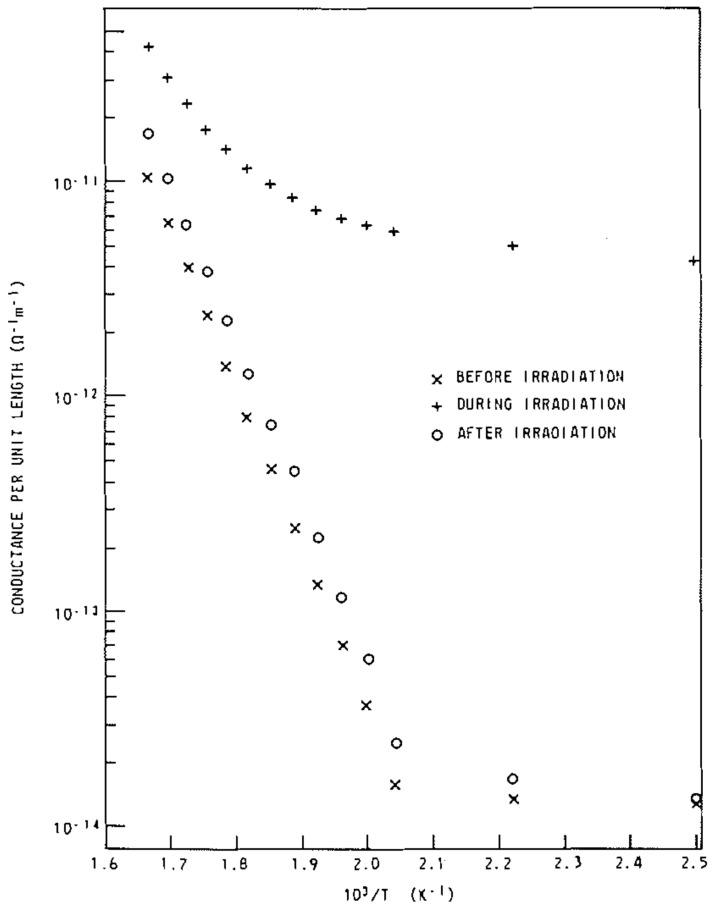
Temperature-dependent photoconductance per unit length of MgO polycrystals before, during, and after γ-ray irradiation [170]. Copyright 1975, Canadian Science Publishing.

**Figure 14 molecules-28-04151-f014:**
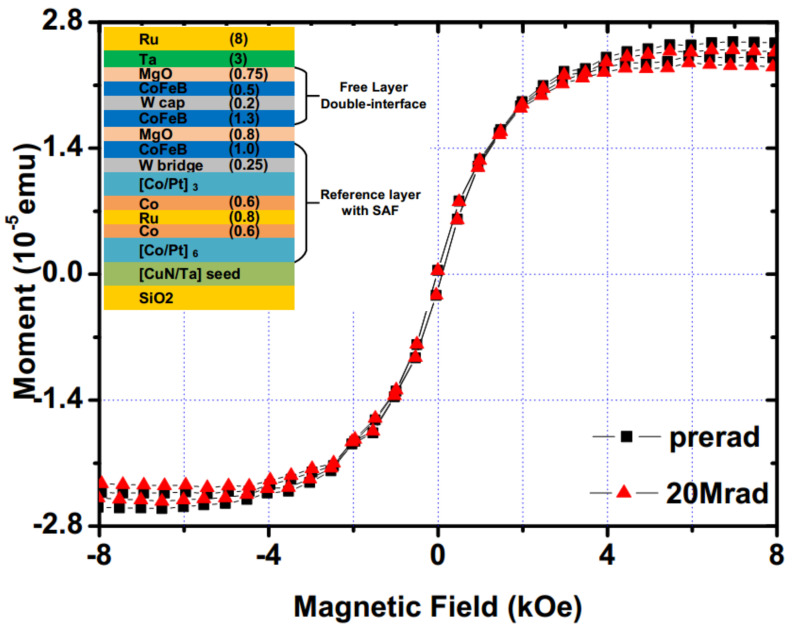
M–H hysteresis loops of MgO-based MTJs measured in an in-plane magnetic field before and after irradiation with a TID of 20 Mrad (SI) [120]. Copyright 2019, IEEE.

**Figure 15 molecules-28-04151-f015:**
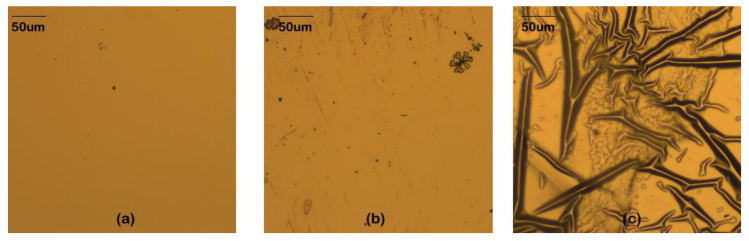
Optical surface images of MgO-based MTJs (**a**) before irradiation, (**b**) after 20 Mrad (Si) irradiation, and (**c**) after 247 Mrad (Si) irradiation [120]. Copyright 2019, IEEE.

**Figure 16 molecules-28-04151-f016:**
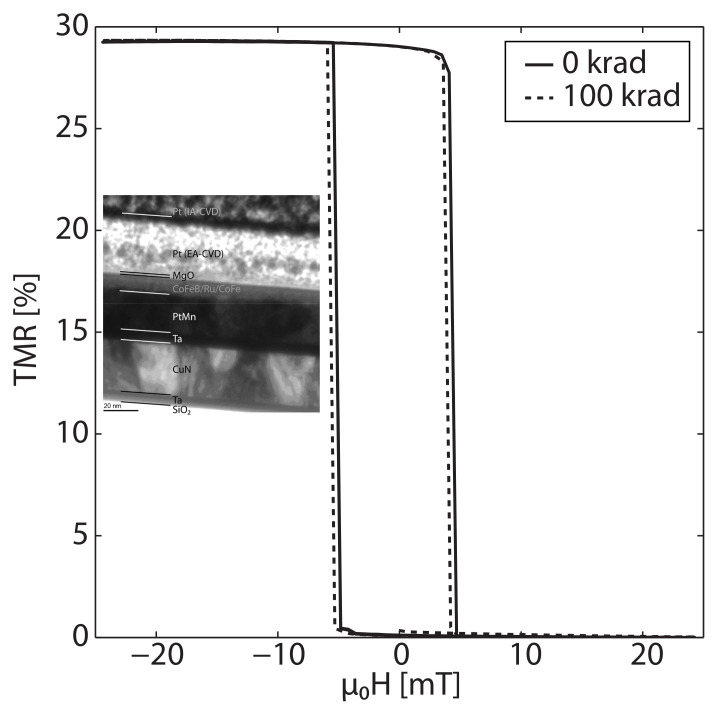
TMR of a single MgO-based MTJ before and after irradiation. Inset: Cross-sectional TEM image [166]. Copyright 2010, International Training Institute for Materials Science. Reproduced with permission [158]. Copyright 2011, IOP Publishing Ltd.

**Figure 17 molecules-28-04151-f017:**
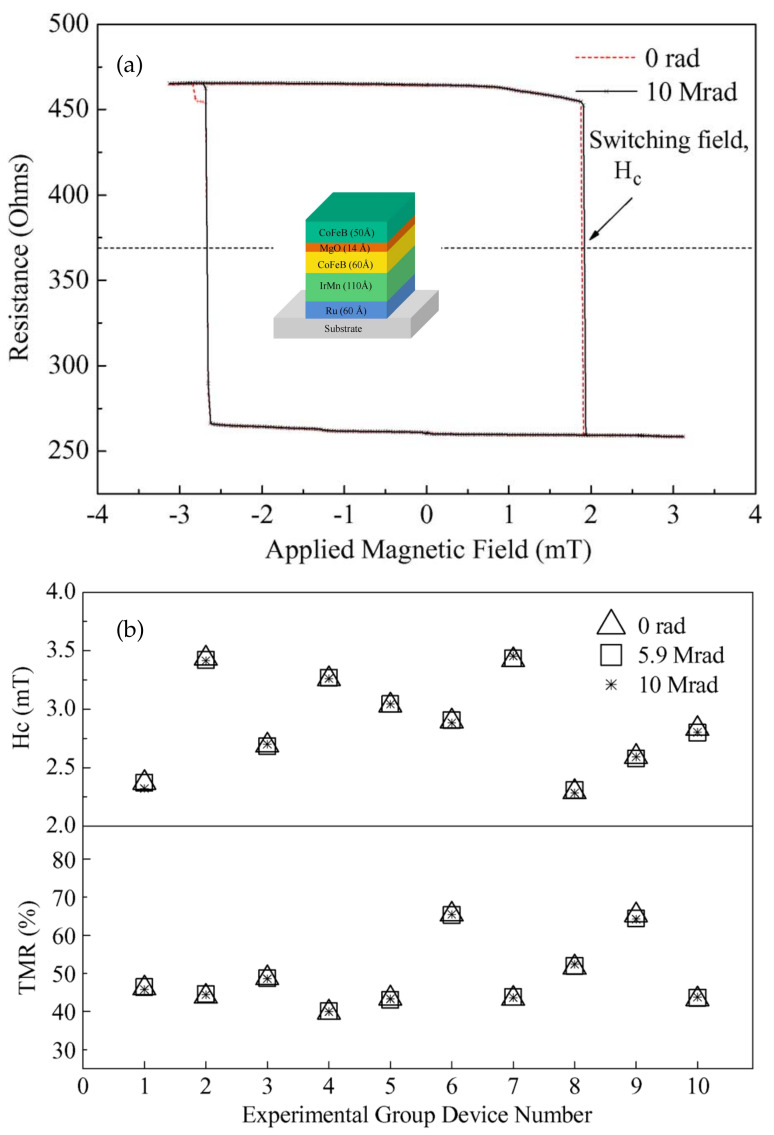
(**a**) Hysteresis loop of a single MgO-based MTJ and (**b**) H${}_{c}$ and TMR of a series of MgO-based MTJs before and after exposure to γ-ray radiation with a dose rate of ∼10rad/s and energy of 1.25MeV. Inset: Illustration of the MTJ stack [152]. Copyright 2012, IEEE.

**Figure 18 molecules-28-04151-f018:**
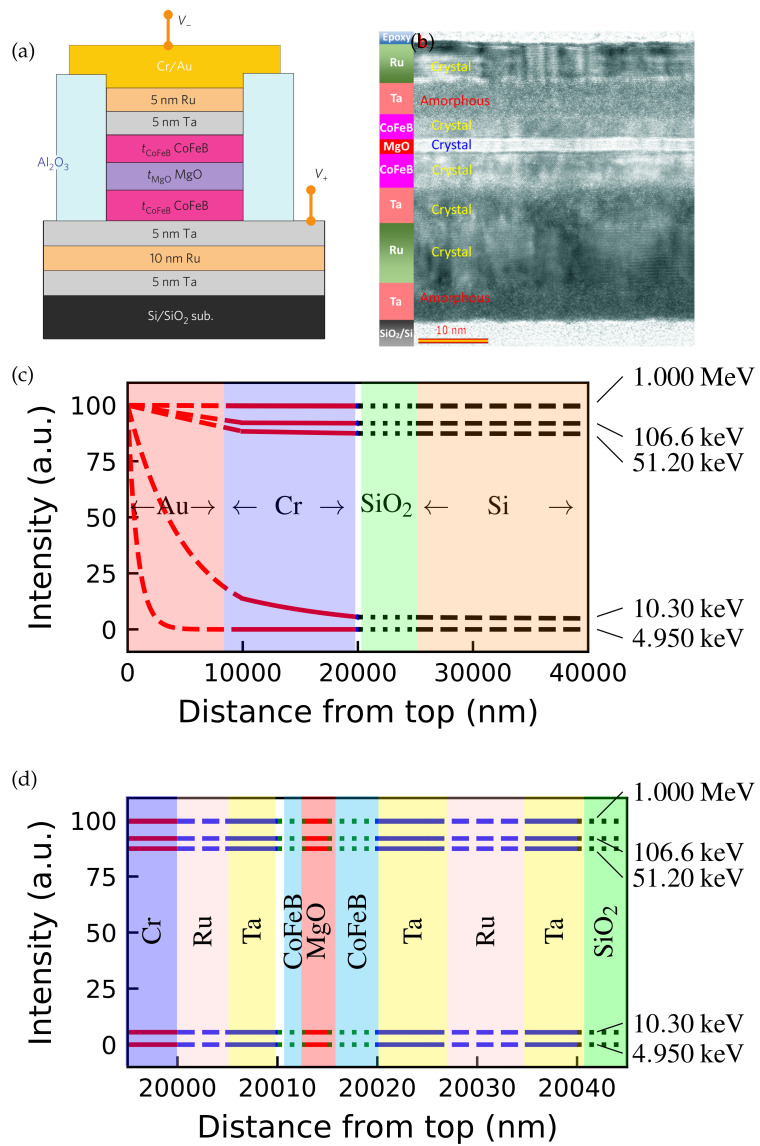
Transmission of electromagnetic radiation through an MTJ device. (**a**) Structure [180] and (**b**) HRTEM cross-sectional image [171] of an MTJ device used for penetration calculations of various types of radiation. Calculated radiation intensity through electrodes (**c**) and sublayers (**d**), including MgO barriers, under various radiation energies. The linear attenuation coefficients of the materials were obtained from https://www.physics.nist.gov (accessed on 17 September 2009). Reproduced with [180] with copyright 2010, Springer Nature. Reproduced with permission [171] with copyright 2016, American Chemical Society.

**Figure 19 molecules-28-04151-f019:**
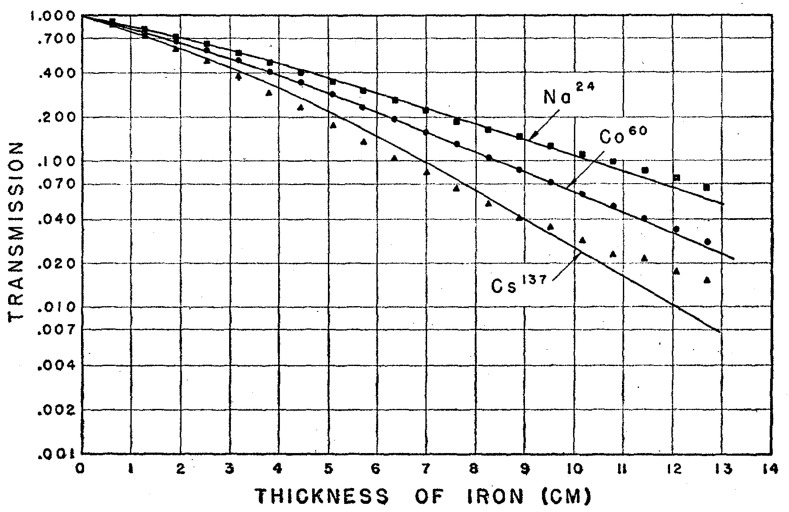
Transmission of γ-radiation through iron. *▲*: ${}^{137}$Cs radiation of 0.66MeV; •: ${}^{60}$Co radiation of 1.17MeV and 1.33MeV; *■*: ${}^{24}$Na radiation of 1.38MeV and 2.76MeV. Reproduced with permission [117]. Copyright 1953, American Physical Society.

**Figure 20 molecules-28-04151-f020:**
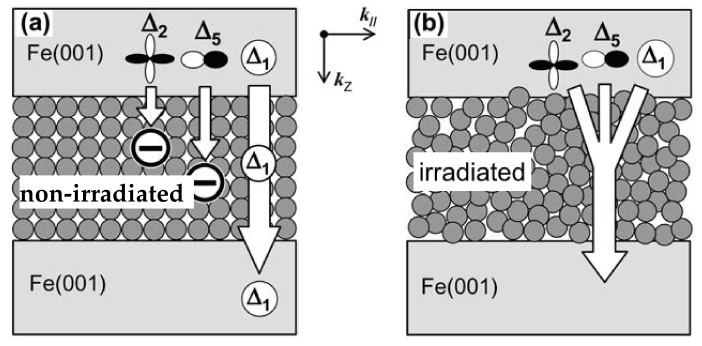
Schematic illustrations of electron tunneling through (**a**) a crystalline barrier and (**b**) an irradiated barrier. $\Delta_{1}:s-p-d$; $\Delta_{2}:d$; $\Delta_{5}:p-d$. Replotted from Ref. [31]. Copyright 2007, IOP Publishing Ltd.

**Figure 21 molecules-28-04151-f021:**
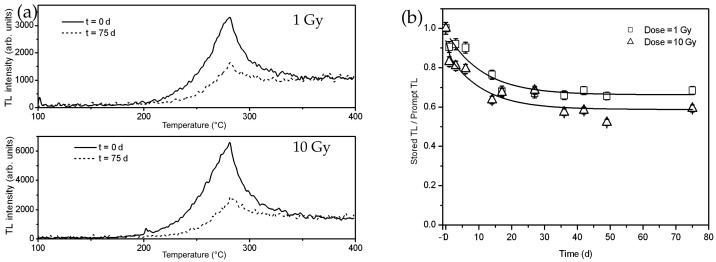
(**a**) Effect of room temperature restoration of irradiated MgO powder measured with a delay period of 75 days (t=75d) and without a delay (t=0d). (**b**) Relative thermoluminescence as a function of restoration time for irradiated MgO. Reproduced with permission [165] with copyright 2009, Taylor & Francis Group.

**Figure 22 molecules-28-04151-f022:**
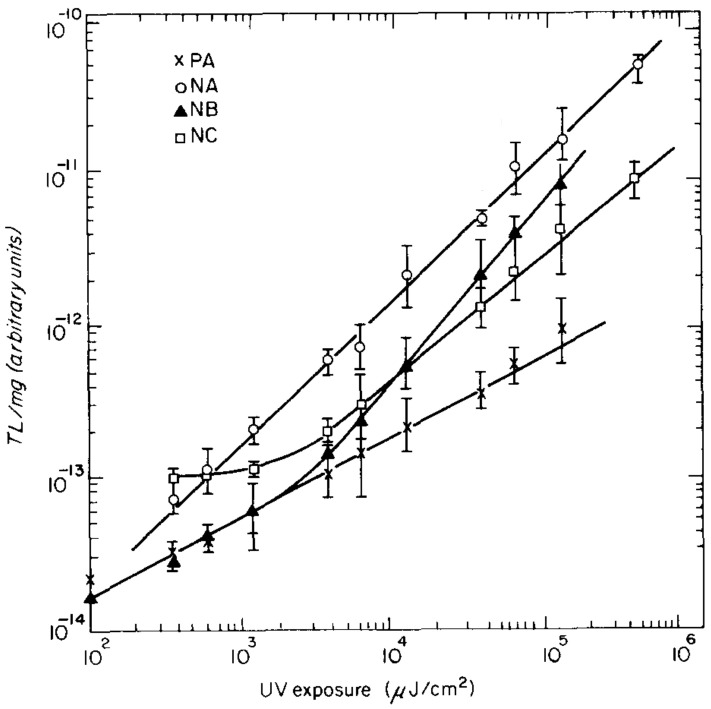
TL response of four MgO crystals as a function of UV exposure at 295 nm. Impurity of PA sample: <0.026; impurity of NA sample: 0.068; impurity of NB sample: 0.082; impurity of NC sample: <0.047. Reproduced with permission [186] with copyright 1976, Am. Assoic. Phys. Med.

**Figure 23 molecules-28-04151-f023:**
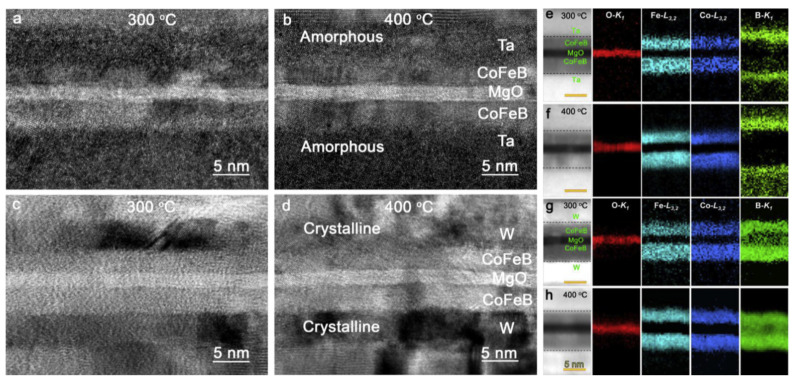
Cross-sectional HRTEM images (**a**–**d**) and ADF-STEM images and corresponding elemental EELS mappings (**e**–**h**) using O-K, Fe-L${}_{3,2}$, Co-L${}_{3,2}$ and B-K ionization edges taken from the Ta/CoFeB/MgO/CoFeB/Ta MTJ (**a**,**b**,**e**,**f**) and W/ CoFeB/MgO/CoFeB MTJ (**c**,**d**,**g**,**h**) at $300{\;}^{\circ }C$ (**a**,**c**,**e**,**g**) and $400{\;}^{\circ }C$ (**b**,**d**,**f**,**h**). Reproduced with permission [193] with copyright 2018, Elsevier.

**Table 1 molecules-28-04151-t001:** Properties of radiation types.

Name	Wavelength	Frequency	Energy
cosmic radiation			up to $10^{20}\;\mathrm{eV}$
γ-ray	<0.01 nm	>30 EHz	>124 keV
X-ray	0.01 nm–10 nm	30 EHz–30 PHz	124.8 eV–124.8 keV
UV	10 nm–400 nm	750 THz–30 PHz	3.12 eV–124.8 eV
visible	400 nm–700 nm	430 THz–750 THz	1.872 eV–3.12 eV
infrared	700 nm–1 mm	300 GHz–430 THz	1.248 meV–1.872 eV
microwave	1 mm–0.1 m	3 GHz–300 GHz	1.248μeV–1.248 meV
radio	>1 m	<3 GHz	<1.248 μeV

KHz: $10^{3}\;\mathrm{Hz}$; MHz: $10^{6}\;\mathrm{Hz}$; GHz: $10^{9}\;\mathrm{Hz}$; THz: $10^{12}\;\mathrm{Hz}$; PHz: $10^{15}\;\mathrm{Hz}$; EHz: $10^{18}\;\mathrm{Hz}$. From Ref. [105] and nasa.gov (accessed on 9 August 2013).

**Table 2 molecules-28-04151-t002:** Some typical radiation sources used in research laboratories.

Sources	Type	Energy	Ref.
Cyclotron	heavy ions	10 MeV	[111,112]
EBIT	heavy ions	tens of keV	[112,113]
Tandem accelerator	particles	20–40 MeV	[112,114]
FIB	gallium ions	30 keV	[112]
Nuclear reactor	neutron	500 MeV	[115]
TEM	electrons	80–200 keV	[116]
SEM	electrons	5 keV–50 keV	[112]
${}^{24}$Na source	γ-rays	2.76MeV, 1.38MeV	[117,118]
${}^{40}$K source	γ-rays	1.46MeV, 1.31MeV	
${}^{60}$Co source	γ-rays	1.33MeV, 1.17MeV	[119,120]
${}^{137}$Cs source	γ-rays	0.66MeV	[121]

TEM: transmission electron microscope; EBIT: electron beam ion trap facility; FIB: focused ion beam.

**Table 3 molecules-28-04151-t003:** Radiation unit and Terms.

Category	Unit	Definition
Activity	Becquerel (Bq) *	activity of a quantity of radioactive material in which one nucleus decays per second (1/s)
	Curie (Ci)	quantity or mass of radium emanation in equilibrium with one gram of radium (element), 1 Ci = $3.7\times 10^{10}\;q$
	Rutherford (Rd)	activity of a quantity of radioactive material in which one million nuclei decay per second, 1 Rd = 1,000,000 Bq
Exposure	Röntgen (R)	quantity of radiation which liberates by ionization one esu ($3.33564\times 10^{10}\;$) of electricity per cm${}^{3}$ of air under normal conditions of temperature and pressure, 1 R = $2.58\times 10^{-4}\;/\mathrm{kg}$
Absorption	Gray (Gy) *	dose of one joule of energy absorbed per kilogram of matter, 1 Gy = 1 J/kg = 100 rad = 10,000 erg/gram
	Radiation absorbed dose (rad)	dose causing 100 ergs of energy to be absorbed by one gram of matter, 1 rad = 0.01 Gy = 100 erg/gram
Absorption	Sievert (Sv) *	equivalent biological effect of the deposit of a joule of radiation energy in a kilogram of human tissue, 1 Sv = 1 J/kg = 100 rem
	Roentgen equivalent man (rem)	unit of health effect of ionizing radiation, 1 rem = 0.010 Sv = 100 erg/gram
Dose		quantity of radiation or energy absorbed
Dose rate		dose delivered per unit of time
Exposure		amount of ionization produced by radiation, the unit is the roentgen (R).

*: SI unit. From epa.gov (accessed on 28 April 2023) and nih.gov (accessed on 1 March 2017).

**Table 4 molecules-28-04151-t004:** Bulk properties of magnesium oxide (MgO) used as a barrier layers in MgO-based MTJs [105].

Physical Property	Values
Space group	Fm$\bar{3}$m, No. 225
Lattice constant	*a* = 4.212Å
Cleavage	<100>
Molar mass	40.3044g/mol
Coordination geometry	Octahedral (Mg${}^{2+}$) and octahedral (O${}^{2-}$)
Density	$3.58\;g/{\mathrm{cm}}^{3}$ ($25{\;}^{\circ }C$)
Solubility in water	0.0062g/L ($0{\;}^{\circ }C$), 0.086g/L ($30{\;}^{\circ }C$)
Melting point	$2852{\;}^{\circ }C$ (3,125 K)
Boiling point	$3600{\;}^{\circ }C$ (3,870 K)
Thermal conductivity	45–60 W/m/K ($25{\;}^{\circ }C$)
Thermal expansion	$138\times 10^{-7}\;{/}^{\circ }C$ ($25{\;}^{\circ }C$)
Heat capacity (C)	37.2J/mol/K ($24{\;}^{\circ }C$)
Std molar entropy ($S_{298}^{{}^{\circ }}$)	26.95J/mol/K
Std enthalpy of formation ($\Delta_{f}H_{298}^{{}^{\circ }}$)	601.6kJ/mol
Gibbs free energy ($\Delta_{f}G_{298}^{{}^{\circ }}$)	−569.3kJ/mol
Electrical conductivity	$10^{-14}\;\mu S/m$ ($24{\;}^{\circ }C$)
Band gap	7.8 eV [123]
Refractive index ($n_{D}$)	1.7355 (λ=0.633μm)
	1.72 (λ=1μm)
Transparency	>92% (λ = 0.25–7 μm)
Thermal stability	up to 700K
Dielectric constant	9.65
Magnetic susceptibility (χ)	$-10.2\times 10^{-6}\;{\mathrm{cm}}^{3}/\mathrm{mol}$

**Table 5 molecules-28-04151-t005:** Physical properties of free/fixed layer materials in MgO-based MTJs.

Property	Fe	Co	(Co,Fe)${}_{80}$B${}_{20}$
space group	Im$\bar{3}$m	P6${}_{3}$/mmc	amorphous [125]
density (g/cm${}^{3}$)	7.87	8.90	7.29
melting point (K)	1811	1768	663–808 * [126]
boiling point (K)	3134	3200	n/a
thermal conductivity (W/m/K)	80.4	100	n/a
electron configuration	[Ar]3d${}^{6}$4s${}^{2}$	[Ar]3d${}^{7}$4s${}^{2}$	n/a
electric conductivity (S/m at RT)	$1.60\times 10^{7}$	$1.04\times 10^{7}$	$10^{6}$–$10^{8}$ [127]
magnetic moment ($\mu_{B}$)	2.2	1.6	2.1–2.5 [128]
Curie temperature (K)	1043	1388	631

from https://www.periodic-table.org (accessed on 11 May 2023) and metglas.com (accessed on 11 May 2023). *: crystallization temperature.

**Table 6 molecules-28-04151-t006:** Cosmic radiation irradiation of MgO-based MTJs.

MTJ Structures	Irradiation Conditions	Results	Ref.
CoFeB/MgO/CoFeB ${}^{†}$	Fe ions, 15 MeV, 400 MeV; Ar, 250 MeV; Kr, 322 MeV; Xe, 454 MeV; Os, 490 MeV	soft errors were detected	[151]
CoFeB/MgO/CoFeB ${}^{\$}$	${}^{60}$Co, γ-ray, 247–475 Mrad, 220rad/s, room temperature	magnetism was destroyed	[120]
CoFeB/MgO/CoFeB ${}^{♯}$	neutron, 0.1 eV–10 MeV, $5\times 10^{10}\;\mathrm{particles}\;/cm^{2}/s$, $2.9\times 10^{15}\;\mathrm{particles}\;cm^{2}$	insensitive	[152]

Numbers in parentheses are nominal thicknesses in nm. ${}^{†}$ Ta(5)/Ru(10)/Ta(5)/Pt(5)/[Co(0.4)Pt(0.4)]${}_{\times 6}$/Co(0.4)/ Ru(0.4)/[Co(0.4)/Pt(0.4)]${}_{\times 2}$/Co(0.4)/Ta(0.3)/CoFeB(1)/MgO/CoFeB(1.5)/Ta(5)/Ru(5); ${}^{\$}$ Ru(8)/Ta(3)/Mg(0.75)/ CoFeB(0.5)/W(0.2)/CoFeB(1.3)/MgO(0.8)/CoFeB(1.0)/W(0.25)/[Co/Pt]${}_{3}$/Co(0.6)/Ru(0.8)/Co(0.6)/[Co/Pt]${}_{6}$/ [CuN/Ta]/Si; ${}^{♯}$ Si/Ru(6)/IrMn(11)/CoFeB(6)/MgO(1.4)/CoFeB(5).

**Table 7 molecules-28-04151-t007:** γ-ray electromagnetic irradiation of MgO-based MTJs.

MTJ Structures	Irradiation Conditions	Results	Ref.
MgO crystals	$3.0\times 10^{6}\;\mathrm{rad}/h$ for 20 min, ${}^{60}$Co, $38{\;}^{\circ }C$, measured within 2 min after irradiation	irradiation produced vacancies	[162]
MgO crystals	γ-ray, 2.1 MeV, up to 10 Mrad, $1.6\times 10^{6}\;\mathrm{rad}/h$, RT	thermal conductivity decreased by half; absorption increased by five times; fully recovered after annealing at $625{\;}^{\circ }C$ for 1 h	[163]
MgO crystals ${}^{⊤}$	γ-ray, 1.25MeV, $10\times 10^{4}\;\mathrm{Gy}$, 0.8 Gy/s, 450K	TSL intensity increased linearly with dose	[160]
MgO crystals ${}^{⊥}$	γ-ray, 1.25MeV, $10\times 10^{4}\;\mathrm{Gy}$, 0.8Gy/s, 450K	TSL intensity was very weakly dependent on dose	[160]
MgO powder	γ-ray (${}^{60}$Co), 0.3Mrads/h, ∼20Mrads, stored at RT for 1 year before measurement	TL changed after irradiation	[164]
MgO powder	γ-ray (${}^{60}$Co), 8.33mGy/s, 1Gy–50kGy	TL changed with dose	[165]
Ag/MgO/Ag ${}^{\nabla }$	γ-ray, 0.662 MeV, up to 32.55 mGy	capacitance increased with dose	[121]
CoFeB films	γ-ray, 1.2 MeV, $2.58\times 10^{5}\;/\mathrm{kg}$, $60{\;}^{\circ }C$	sensitive to γ-ray irradiation	[118]
MgO/CoFeB ${}^{§}$	γ-ray, 100 kRad	no noticeable change in magnetic properties	[166]
CoFeB/MgO/CoFeB	${}^{60}$Co, γ-ray, 1 Mrad	no effect	[119]
CoFeB/MgO/CoFeB ${}^{¶}$	${}^{60}$Co, γ-ray, 10 Mrad, 9.78 rad/min	highly tolerant of γ-ray radiation	[152]
CoFeB/MgO/CoFeB ${}^{‡}$	${}^{60}$Co, γ-ray, below 20 Mrad, 220rad/s, RT	coercivity increased with irradiation while saturation magnetization was not affected	[120]

Numbers in parentheses are nominal thicknesses in nm. ${}^{⊤}$ MgO crystals with OH${}^{-}$ impurity of $\left(4.7-4.9\right)\times 10^{17}\;/{\mathrm{cm}}^{3}$. ${}^{⊥}$ MgO crystals without OH${}^{-}$ impurity. ${}^{\nabla }$ Ag/MgO thick film/Ag. Grain size of MgO particles: 0.5–1.0 μm. Ag was electrode. ${}^{§}$ Ru(7)/Ta(10)/Co${}_{60}$Fe${}_{20}$B${}_{20}$(3)/Mg(0.3)/MgO(1.1)/Co${}_{60}$Fe${}_{20}$B${}_{20}$(3)/ Ru(0.8)/Co${}_{70}$Fe${}_{30}$(2.5)/PtMn(20)/Ta(5)/CuN(30)/Ta(5). ${}^{¶}$ CoFeB(5)/MgO(1.4)/CoFeB(6)/IrMn(11)/Ru(6). ${}^{‡}$ [Co(0.5)/Pt(0.2)]${}_{\times 6}$/Co(0.6)/Ru(0.8)/Co(0.6)/[Co(0.5)/Pt(0.2)]${}_{\times 3}$/W(0.25)/CoFeB(1.0)/MgO(0.8)/CoFeB(1.3)/ W(0.2)/CoFeB(0.5)/MgO(0.75)/Ta(3.0)/Ru(8.0). RT: room temperature; TSL: thermally stimulated luminescence.

## Data Availability

Not applicable.

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
