# Peer review of "Electromagnetic Radiation Effects on MgO-Based Magnetic Tunnel Junctions: A Review"

_molecules, 2023, doi:10.3390/molecules28104151_

Round 1
Reviewer 1 Report
In this paper, the authors have reviewed the effects of electromagnetic Irradiation on MgO-based magnetic tunnel junctions. Current research is reviewed to explore the origin of irradiation tolerance in exposed MTJs.
To me, these authors have clearly missed the primary object of this Special Issue. For instance, their only citation (or reference) to anything remotely related to "energy" (and not even energy conversion or energy storage) is the allusion to "sensing" (see Ref [51]). The article cited in Ref [51] focuses on magnetic tunnel junction-based microwave detection and its ability to detect microwave frequencies. So, how exactly is this article related to energy conversion and storage?
They should modify their aims of this review paper to indicate clearly how this review aligns with the stated aims of this Special Issue, i.e., "to publish papers related to synthesis and novel process methods, structures and properties, development and applications, and the improvement of energy nanomaterials in terms of energy conversion and storage.” In fact, they must answer the following question within this review paper:" what role does an irradiated MTJ play in "energy conversion" and "storage"?
The Introduction and focus of the discussions presented in this review article require substantial revision to align it with the stated priority of the Special Issue it is submitted to. This is important because what the authors have presented here is entirely out of sync with the focus of the special issue, they are submitting this article.
I can not recommend this article for publication in the stated Special Issue because the materials discussed herein do not fit the indicated theme. This review article does not in any way "aid the development of novel nanomaterials and promote an improved understanding of the related working mechanisms", as stated here: https://www.mdpi.com/journal/molecules/special_issues/energetic_nanomaterials.
Author Response
We express our gratitude to the reviewer for the valuable feedback. In response to the suggestion, we have incorporated a new subsection on energy harvesting and storage, which aligns with the theme of the issue. Additionally, we have included more references to strengthen the article. The revised version highlights the new additions in blue.
The manuscript's English has been enhanced.

Reviewer 2 Report
Although this review article covers most of important application aspects of MgO based magnetic tunnel junctions under radiation, the coverage of the topics is mostly superficial with almost no in-depth discussions. In a sense, it is not much more than a collection of relevant references in the field. This may be useful to some researchers, but because of the lack of in-depth discussion and absence of any analysis of the field, the impact of this work will be very limited.
The authors seem to be content with citing the references within each topic being reviewed, without any effort to connect the literature into the narrative of the review. Thus the reader will not be able to gain much insight from the review.
There are also serious deficiencies in the coverage. The authors recognize the importance of the interface but only did a cursory coverage with a single paragraph in section 3.1.3, and blame it on the lack of literature on gamma-irradiated MgO junctions. A serious reviewer in such cases would have searched related fields to find work that can provide insight on this topic, and connect them to the topic of interest. Furthermore, a quick google search yielded dozens of papers on the topic. I would suggest that the authors re-review this.
I cannot find any discussion of radiation induced diffusion, and how this would degrade the performance of the MTJ. The issue of diffusion in MTJs is an important topic and has a large literature.
Some technical issues listed below:
line 75-76: electrons travel with their spins oriented perpendicularly to ferromagnetic layers
"perpendicularly" misplaced.
line 113-114: MR ratios higher than 150 % at room temperature was predicted
First-principles works did not predict room temperature MR. Low temperature MR predicted was several thousand percent.
line 160:
It is generally accepted that the Julliére’s model can explain the insulating-barrier-based TMR effect.
This statement is debatable and also contradicts the following paragraphs. The Julliere model may only be applicable to amorphous barrier layers when there is no symmetry filtering effect. The inadequacy of the Julliere model in general is well-documented. For example, see MacLaren et al, Phys. Rev. B 56, 11827 (1997).
Author Response
The authors seem to be content with citing the references within each topic being reviewed, without any effort to connect the literature into the narrative of the review. Thus the reader will not be able to gain much insight from the review.
Rep: We would like to express our gratitude to the reviewer for the valuable suggestions. As per the suggestions, we have made several revisions to our manuscript. We have rephrased many sentences and established connections between relevant literature in specific topics. Our aim is to provide the reader with a comprehensive understanding of the concepts of irradiation and magnetic tunnel junctions (MTJs), followed by an overview of the scientific research on the effects of irradiation on MTJs. Finally, we have provided theoretical analyses and explanations of the reported phenomena. We believe that these revisions have helped to present a clear and concise picture of the irradiation of MTJs.
There are also serious deficiencies in the coverage. The authors recognize the importance of the interface but only did a cursory coverage with a single paragraph in section 3.1.3, and blame it on the lack of literature on gamma-irradiated MgO junctions. A serious reviewer in such cases would have searched related fields to find work that can provide insight on this topic, and connect them to the topic of interest. Furthermore, a quick google search yielded dozens of papers on the topic. I would suggest that the authors re-review this.
Rep: Thank the review for the comments. We searched online and added more information, as highlighted in the reversed manuscript. Some paragraphs are added as below:
Recent in-situ experiments discovered that the uniaxial magnetic anisotropy decreased systematically with increasing annealing temperature [ 167 ]. Specifically, the MgO/FeCoB/MgO layers becomes isotropic after annealing at 450 ◦C. The asymmetry at the interfaces was explained by the diffusion of boron from the FeCoB interface layer into the adjacent MgO layer. Electronic structures of MgO/Fe interfaces have been investigated [ 168 ]. It is believed that Fe 3d -O 2p hybridization and distortion of the Fe film play important roles in magnetic anisotropy at the MgO/Fe interface.
A study by Ueda et al. has explored the electronic structures of MgO/Fe interfaces. Their findings suggest that the magnetic anisotropy at the MgO/Fe interface is influenced by the hybridization between Fe 3d and O 2p orbitals, as well as by the distortion of the Fe film.
I cannot find any discussion of radiation induced diffusion, and how this would degrade the performance of the MTJ. The issue of diffusion in MTJs is an important topic and has a large literature.
Rep: Section 4.3, titled "Infrared Irradiation and Thermal Annealing" contains a discussion on radiation-induced diffusion. Since radiation typically induces heat, the discussion was included in this section. We also included an additional paragraph to establish a link between the thermal effect and irradiation-induced diffusion.
Some technical issues listed below:
line 75-76: electrons travel with their spins oriented perpendicularly to ferromagnetic layers
"perpendicularly" misplaced.
Rep: Thanks to the reviewer for the careful reading of the manuscript. The typo was corrected.
line 113-114: MR ratios higher than 150 % at room temperature was predicted
First-principles works did not predict room temperature MR. Low temperature MR predicted was several thousand percent.
Rep: Thank the reviewer. We re-checked the literature and the sentence was updated.
line 160:
It is generally accepted that the Julliére’s model can explain the insulating-barrier-based TMR effect.
This statement is debatable and also contradicts the following paragraphs. The Julliere model may only be applicable to amorphous barrier layers when there is no symmetry filtering effect. The inadequacy of the Julliere model in general is well-documented. For example, see MacLaren et al, Phys. Rev. B 56, 11827 (1997).
Rep: Thank the reviewer to point out the problem. The publication was cited. The sentence "It is generally accepted that the Julliére’s model can explain the insulating-barrier-based TMR effect." was changed to:
According to a report in 1997 [ 121 ], the Julliere model is not a precise representation of the magnetoconductance exhibited by free electrons tunneling through a barrier. Instead, in the case of thick barriers, Slonczewski’s model may offer a more accurate approximation. However, despite this limitation, we will use the Julliere model here to illustrate the creation of an amorphous state in barrier layers induced by irradiation, as it provides a simpler representation.

Round 2
Reviewer 1 Report
The authors have revised their manuscript in line with the points I made in my first report. In my view, doing so has made their review paper to be improved substantially, and therefore suitable for publication in the Special Issue. However, they must address the following specific minor issues before the paper can be published:
(1). It is strange that the entire Section 1.3.3 on “Radiation Units” has no references (or citations) to standard sources in any manner whatsoever. The authors' allusion to standard definitions (as listed in Table 3) is insufficient for this purpose insofar as no citations are also listed within Table 3. The authors must provide suitable citations within this subsection.
(2). Provide appropriate citations for the variations of “the intensity of electromagnetic radiation inside MTJs”, wherein empirical data showing that it “decreases exponentially from MTJ’s surfaces, as described by the equation based on the Beer-Lambert law”, as stated in lines 827 and 828.
(3). With a full listing of appropriate citations, the authors should first define what they mean by “temporal defects” on Page 34 before writing anything else under it. This is important because that term is a non-standard in the spectroscopy of defects in solids. After addressing that point, they should read again their sentence in line 390 and correct it.
(4). The caption used for Section 4 is vague, and tells the reader nothing meaningful. What is the physical meaning of “Other Irradiation” in “Effects of Other Irradiation”? Otherwise, the authors must state clearly within section “1.3. Irradiation” what types of irradiation of the MTJ will be covered in the review.
(5). Merge the single-sentence paragraph in Line 991 with the previous paragraph.
Author Response
(1). It is strange that the entire Section 1.3.3 on ?Radiation Units? has no references (or citations) to standard sources in any manner whatsoever. The authors' allusion to standard definitions (as listed in Table 3) is insufficient for this purpose insofar as no citations are also listed within Table 3. The authors must provide suitable citations within this subsection.
Rep: Thank the reviewer for the excellent comments. We have made the necessary revisions and added references to Section 1.3.3 and Table 3. Additionally, we have added several references to Table 2. Moreover, we have included some references to Sections 1.3.1 and 1.3.2 as well. Please let us know if there is anything else we can do to improve the manuscript.
(2). Provide appropriate citations for the variations of ?the intensity of electromagnetic radiation inside MTJs?, wherein empirical data showing that it ?decreases exponentially from MTJ?s surfaces, as described by the equation based on the Beer-Lambert law?, as stated in lines 827 and 828.
Rep: Thanks again for the comment. We performed the calculations ourselves using the structure of MTJ, and the results have not been published yet. In addition, we have included the references related to the Beer-Lambert law, which describes X-ray and gamma-ray absorption in solid materials, as well as a recently published paper on how to calculate the intensity of multilayers. We hope these additions address your concerns and improve the manuscript.
(3). With a full listing of appropriate citations, the authors should first define what they mean by ?temporal defects? on Page 34 before writing anything else under it. This is important because that term is a non-standard in the spectroscopy of defects in solids. After addressing that point, they should read again their sentence in line 390 and correct it.
Rep: Thanks for the comment. We have made the necessary changes and replaced the word "temporary" with "intermittent," which is more commonly used in defects. We have also updated the sentences related to "temporary defects." We appreciate your comments and hope these revisions could improve the quality of the manuscript.
(4). The caption used for Section 4 is vague, and tells the reader nothing meaningful. What is the physical meaning of ?Other Irradiation? in ?Effects of Other Irradiation?? Otherwise, the authors must state clearly within section ?1.3. Irradiation? what types of irradiation of the MTJ will be covered in the review.
Rep: Thank the reviewer again for the comments. We have updated the title of Section 4 to "low-energy radiation" as suggested. Furthermore, we have added a paragraph after the section title to clarify that electromagnetic waves with wavelengths longer than gamma-rays are commonly referred to as lower energy waves, including X-rays, ultraviolet radiation (UV), visible light, infrared radiation, microwaves, and radio waves. These waves typically have less energy than gamma-rays and are typically classified as non-ionizing radiation, with the exception of X-rays.
Additionally, we have made some minor adjustments to the wording in the abstract to improve its clarity and conciseness.
(5). Merge the single-sentence paragraph in Line 991 with the previous paragraph.
Rep: Did. Thank you.
Some typos in the manuscript are corrected.
Reviewer 2 Report
This version is significantly improved. The authors have addressed most of the criticism from the previous review. The paper is almost ready for publication. However, there remains a few minor issues that still need to be addressed.
1. There is significant repetition in the descriptions of TMR history in lines 90-99 and in lines 116-142. They should be consolidated into a single paragraph.
2. Line 84-85, the sentence "The tunnel magnetoresistance (TMR) phenomenon is an extension of GMR in which the electrons travel with their spins oriented parallelly to the ferromagnetic layers across a thin non-metallic tunneling barrier" is still problematic. Why do the electron spins need to be parallel (or perpendicular) to the ferromagnetic layers? This is not necessary for TMR. It is not clear to me what the authors are trying to express.
3. Line 401. Jullere model is more likely to be correct for amorphous barrier than crystalline barrier. It would be better if this point is made more explicit.
4. Line 762. What is the consequence of boron diffusion into the MgO barrier? This was studied from first-principles theory by Bai et al Phys. Rev. B 87, 014114 (2013). Have there been follow-up experiments? For a good review article, the authors need to dig a little deeper into the literature.
Author Response
1. There is significant repetition in the descriptions of TMR history in lines 90-99 and in lines 116-142. They should be consolidated into a single paragraph.
Rep: Thank the reviewer for the suggestion. We have combined the three paragraphs and rewritten them into a single paragraph. We hope this makes the section more concise and easier to read.
2. Line 84-85, the sentence "The tunnel magnetoresistance (TMR) phenomenon is an extension of GMR in which the electrons travel with their spins oriented parallelly to the ferromagnetic layers across a thin non-metallic tunneling barrier" is still problematic. Why do the electron spins need to be parallel (or perpendicular) to the ferromagnetic layers? This is not necessary for TMR. It is not clear to me what the authors are trying to express.
Rep: Thank the comments. The paragraph was changed to:
"Tunnel magnetoresistance (TMR) can be considered an extension of giant magnetoresistance (GMR) due to their similarities in electrical resistance changes of magnetic multilayer structures by aligning the magnetic moments of adjacent layers. Different from GMR, TMR employs a thin insulating layer as a tunneling barrier between magnetic layers, resulting in quantum mechanical electron tunneling across the barrier and leading to more significant changes in electrical resistance compared to GMR devices."
3. Line 401. Jullere model is more likely to be correct for amorphous barrier than crystalline barrier. It would be better if this point is made more explicit.
Rep: Thank the reviewer for the comment. We agree with the reviewer that the Julliere model is commonly used to explain amorphous barrier MTJs. In our case, as the crystalline barriers may be amorphized by irradiation, we employed this model to explain how the radiation-induced amorphous state affects the performance of MTJs. To clarify the reason for using the Julliere model in this manuscript, we have revised and added some sentences to the manuscript. We hope these changes help to better explain our approach and address the reviewer's concerns.
4. Line 762. What is the consequence of boron diffusion into the MgO barrier? This was studied from first-principles theory by Bai et al Phys. Rev. B 87, 014114 (2013). Have there been follow-up experiments? For a good review article, the authors need to dig a little deeper into the literature.
Rep: Thank the reviewer for the comment. We have cited the theoretical paper along with an experimental publication in the revised manuscript. We have also added a paragraph to discuss the impact of boron diffusion on MTJ performance.
Furthermore, we have included experimental work on the interface and added another figure to the manuscript. We hope that these changes address your concerns and improve the clarity and completeness of our paper.
Some typos in the manuscript are corrected.